# Toward Protein Docking-oriented De Novo Ligand Design via Gradient Inversion

## Abstract

De novo ligand design is a fundamental task that seeks to generate protein or molecule candidates that can effectively dock with protein receptors and achieve strong binding affinity entirely from scratch. It holds paramount significance for a wide spectrum of biomedical applications. However, most existing studies are constrained by the **Pseudo De Novo**, **Limited Docking Modeling**, and **Inflexible Ligand Type**. To address these issues, we propose MagicDock, a forward-looking framework grounded in the progressive pipeline and differentiable surface modeling. (1) We adopt a well-designed gradient inversion framework. To begin with, general docking knowledge of receptors and ligands is incorporated into the backbone model. Subsequently, the docking knowledge is instantiated as reverse gradient flows by binding prediction, which iteratively guide the de novo generation of ligands. (2) We emphasize differentiable surface modeling in the docking process, leveraging learnable 3D point-cloud representations to precisely capture binding details, thereby ensuring that the generated ligands preserve docking validity through direct and interpretable spatial fingerprints. (3) We introduce customized designs for different ligand types and integrate them into a unified gradient inversion framework with flexible triggers, thereby ensuring broad applicability. Moreover, we provide rigorous theoretical guarantees for each component of MagicDock. Extensive experiments across 9 scenarios demonstrate that MagicDock achieves average improvements of 27.1% and 11.7% over SOTA baselines specialized for protein or molecule ligand design, respectively.

## 1 Introduction

De novo ligand design is a cornerstone of bioengineering, centered on the creation of ligands—such as proteins and molecules—with strong binding affinity to target protein receptors, thereby forming highly stable complexes with substantial biological potential. Traditionally, ligand design has relied on energy optimization techniques ( Adolf-Bryfogle et al. (2018)). Recent advances in deep learning have transformed the field by introducing powerful data-driven methods, substantially enhancing generative capabilities ( Evans et al. (2021); Gu et al. (2024)). Despite their effectiveness, existing methods still face inherent limitations, as shown in Fig. 1. **(1) Pseudo De Novo**. They inherently remain dependent on prior knowledge. Specifically, some antibody design methods depend heavily on predefined structural templates—such as fixed frameworks and conserved CDR regions excluding CDR-H3—thereby restricting the design space around the most critical binding region for optimization while sacrificing the capacity to generate antibodies de novo ( Zhou et al. (2024)).

Furthermore, progress toward fully de novo ligand design remains hindered by two additional critical issues. **(2) Limited Docking Modeling.** Current methods typically employ indirect docking representation methods to capture docking performance (such as energy functions assessing docking tightness through residue-level biophysical terms) ( Zhou et al. (2024)), without explicitly considering spatial docking information and protein surface information (which may miss key docking recognition information), leading to the inability to ensure robust biological relevance of generated ligands. **(3) Inflexible Ligand Type.** Most existing approaches are narrowly tailored to specific ligand types like protein or molecule ( Luo et al. (2022); Guo et al. (2021)), which severely limits their versatility and applicability across diverse molecular categories. Collectively, these challenges hinder the development of a comprehensive and generalizable framework for ligand design, thereby constraining progress in drug discovery and biomolecular engineering.

Figure 1: Comparison between current works and MagicDock. This figure describes the three limitations of the existing methods and presents the framework of MagicDock, which achieves authentic de novo, biological significance and cross-category generality.

To address the above critical issues, we propose MagicDock, a forward-looking framework rooted in differentiable surface modeling for de novo ligand design. Our approach is designed with three core innovations. (1) We introduce a **well-designed gradient inversion framework**. To begin with, the backbone model learns general molecular-level knowledge from large protein and molecule datasets. Subsequently, the model acquires docking-specific knowledge by refining its encoders through three progressively structured downstream tasks. By integrating general knowledge and specific docking knowledge into the model, this inversion framework can directly utilize the gradient information contained in the model to guide the generation of ligands from scratch. (2) We emphasize **differentiable surface modeling through learnable 3D point-cloud representations**, enabling the framework to capture fine-grained spatial binding fingerprints with interpretability. This ensures that the generated ligands incorporate spatial and surface information, enabling the ligands to perfectly align with the receptors both geometrically and biologically. (3) We design **customized modules for different ligand types**, which are seamlessly integrated into a unified gradient inversion framework with flexible triggering mechanisms. This design enables convenient switching of different ligand types in generation, improving both efficiency and flexibility in ligand generation. Importantly, the validity and efficiency of our method is supported by rigorous theoretical guarantees in Sec. 4 such as SE(3)-equivariance across stages and superiority over other methods, ensuring both methodological soundness and practical reliability.

**Our contributions.** (1) *New Perspective*: We introduce docking-oriented inversion as an innovative framework for de novo ligand design, addressing challenges in genuineness, biological significance, and cross-category generality. (2) *New Framework*: We introduce a novel inversion framework that leverages gradient-based optimization, starting from surface point cloud modeling of proteins and ligands, through docking-oriented knowledge injection process, to enable inversion for generating de novo ligands directly within the receptor's binding pocket. (3) *New Method*: We propose a differentiable data structure for seamless gradient flow, integrated with flexible triggers, ensuring flexibility and biological relevance in ligand generation. (4) *SOTA Performance*: Compared across 9 scenarios, **MagicDock** achieves state-of-the-art performance in designing high-affinity ligands, having an average improvement of over 70% in protein ligand design and over 60% in molecule ligand design compared with other baselines.

## 2 PRELIMINARIES & RELATED WORKS

### 2.1 NOTATIONS AND PROBLEM FORMULATION

We adopt a docking-oriented docking strategy, as surface point clouds capture fine-grained structural cues for molecular recognition and docking data provide binding compatibility. Both protein and small-molecule ligands are represented as 3D surface point clouds $\mathcal{P} = \{\mathbf{f}_i\}_{i=1}^{N}$, where $\mathbf{f}_i \in \mathbb{R}^d$ encodes chemical, atomic and geometric features.

Ligand generation proceeds in two stages: iterative refinement by a type-specific generator $G_{\text{type}}$ guided by a pre-trained model $M_\theta$, followed by generation via docking energy minimization:

$$\hat{\mathcal{P}}_{\text{lig}} = \lim_{t \to T} G_{\text{type}}\left(\mathcal{P}_{\text{lig}}^{(t)}, M_\theta\right), \quad \mathcal{P}_{\text{lig}}^* = \arg\min_{\hat{\mathcal{P}}_{\text{lig}}} E_{\text{dock}}\left(\mathcal{P}_{\text{rec}}, \hat{\mathcal{P}}_{\text{lig}}\right), \tag{1}$$

where $t$ denotes the refinement step. $G_{\text{type}}$ incorporates domain-specific generating constraints, emphasizing ring and valency rules for small molecules and residue/conformation features for proteins.

## 2.2 RELATED WORKS

Existing methods for protein and molecular ligand design can be categorized by how they *couple generation with optimization*, i.e., the extent to which candidates are refined toward biochemical objectives. We summarize them into four paradigms:

① **Decoupled Paradigms.** Traditional pipelines treat representation, generation, and optimization as separate modules. Classical docking methods such as ZDOCK ( Chen et al. (2010)), RosettaDock ( Lyskov & Gray (2008)), AutoDock ( Morris et al. (2008)), and Glide ( Halgren et al. (2004)) encode ligands into atomic/residue descriptors, followed by independent search and scoring. Early generative models like sequence-based RNNs ( Liu et al. (2020)) also rely on post hoc optimization. These approaches are interpretable but loosely connected to task objectives.

② **Implicitly Coupled Paradigms.** Diffusion-based approaches embed optimization into sampling. For proteins, DiffAb ( Luo et al. (2022)) and HSRN ( Jin et al. (2022)) refine CDR loops with SE(3)-equivariant models. For ligands, DiffDock ( Corso et al. (2022)) and GeoDiff ( Xu et al. (2022)) integrate docking or $\Delta G$ signals into denoising, while Pocket2Mol ( Peng et al. (2022)) and TankBind ( Lu et al. (2022)) further condition on receptor pockets. Although effective, these rely on handcrafted schedules and stochastic trajectories, limiting efficiency and de novo completeness.

③ **Surrogate- or Heuristic-coupled Paradigms.** Another line combines generation with heuristic optimization or surrogate models. For proteins, reinforcement learning (ABDPO ( Zhou et al. (2024))) and memory-augmented models like dyMEAN ( Kong et al. (2023)) incorporate docking rewards. For ligands, reinforcement learning ( Gottipati et al. (2020)), evolutionary strategies ( Chen et al. (2021)), and Bayesian optimization ( Moss et al. (2020)) guide fragment assembly or mutation. Frameworks such as DockStream ( Guo et al. (2021)), ALIDIFF ( Gu et al. (2024)), and DRUGFLOW ( Schneuing et al. (2025)) embed chemical, geometric, or physical priors. These methods are flexible but computationally costly.

④ **Latent-gradient coupling Paradigm (Ours).** Inversion differs by explicitly treating generation as optimization ( Niu et al. (2025);Qiu et al. (2024); Bergues et al. (2025)). Structures are refined via gradient updates in latent space, guided by task-specific losses and domain constraints, without stochastic schedules or heuristic surrogates. More details of Sec. 2.2 are applied in Appendix B.

## 2.3 THE INVERSION FRAMEWORK

As an emerging generative framework, Inversion differs fundamentally from mainstream generative frameworks like Diffusion(Ho et al. (2020); Song et al. (2022)) and Flow-matching (Lipman et al. (2023); Liu et al. (2022)), as it employs gradient-based refinement to iteratively adjust structures toward task-specific goals. Its advantages can be summarized as follows: ① Generality. Traditional frameworks often rely on domain-specific priors or handcrafted surrogates, limiting adaptability. Inversion only requires differentiable embeddings and universal gradient updates, enabling one architecture to generalize once the backbone encodes domain knowledge. ② Efficiency. Mainstream methods incur redundant stochastic trajectories or surrogate solvers, leading to high cost and limited efficiency. Inversion directly couples generation and optimization via gradients, achieving higher information efficiency and a stronger theoretical ceiling (Appendix E.3 and E.5). ③ Modularity and Scalability. Existing approaches are often entangled with backbone designs, hindering transfer of pretrained improvements. Inversion remains orthogonal to the backbone, so better representations directly translate into stronger gradient guidance. Concretely, we divide the inversion framework into two stages:

❶ **Knowledge-infused Pre-training Stage.** A model $M$ is first pre-trained to transform structured data $\mathbf{S}_{\text{pre}}$ (e.g., molecular graphs or 3D point clouds) into comprehensive latent embeddings. This critical stage captures domain-specific features (e.g., chemical and geometric properties) from extensive datasets, ensuring that the learned representations provide robust and meaningful gradients for downstream refined optimization. The training objective can be formulated as:

$$\theta^* = \arg\min_{\theta} \ \mathcal{L}_{\text{pre}} \left( M_{\theta}(\mathbf{S}_{\text{pre}}), \mathbf{Y} \right), \tag{2}$$

where $\theta$ denotes model parameters, $\mathbf{Y}$ represents supervision signals (or self-supervised targets), and $\mathcal{L}_{\text{pre}}$ is a specialized domain-aware loss function designed for robust optimization.

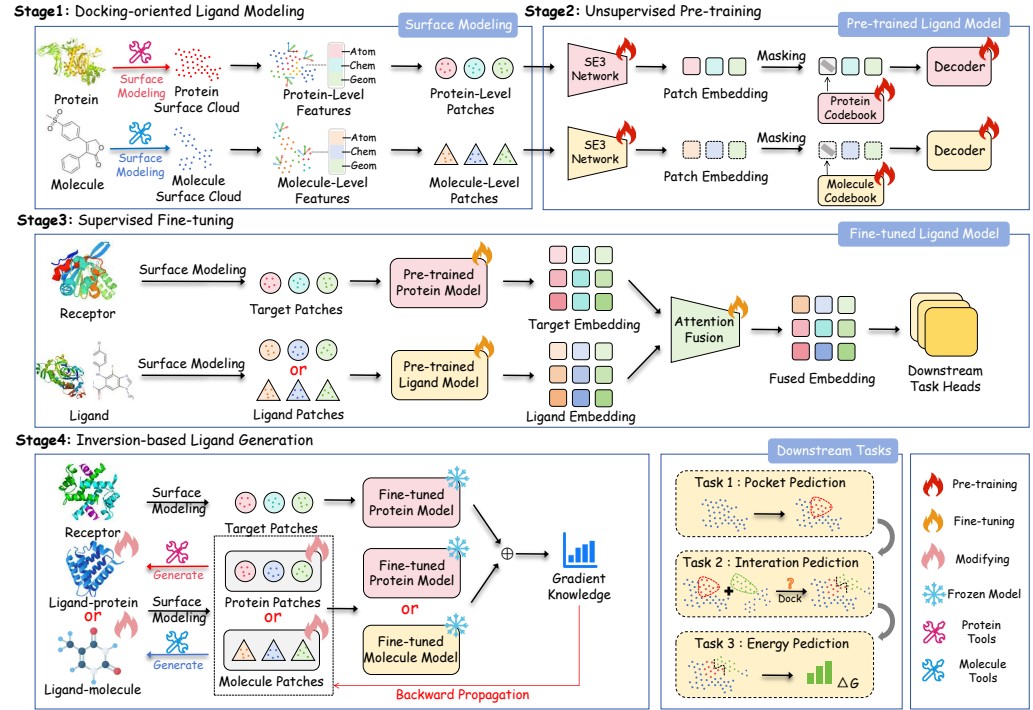

Figure 2: The overview of MagicDock Framework.

❷ **Gradient-driven Inversion Stage.** Once pre-trained, the model iteratively refines structures $\mathbf{S}$ via gradient-based inversion. Starting from an initial $\mathbf{S}_0$, the updates follow:

$$\mathbf{S}_{t+1} = \mathbf{S}_t - \eta \nabla_{\mathbf{S}_t} \mathcal{L}(\mathbf{S}_t, C_{\text{domain}}), \tag{3}$$

where $\mathcal{L}$ is a specialized task-specific objective, $\nabla_{\mathbf{S}_t}$ represents gradients with respect to the current structure, and $C_{\text{domain}}$ rigorously enforces structural or biochemical restrictions.

In our framework, $\mathbf{S}_t$ corresponds to ligand point clouds $\mathcal{P}_{\text{lig}} = \{\mathbf{f}_i\}_{i=1}^N$, which are optimized directly with gradient signals. By applying different domain-specific generating constraints $C_{\text{domain}}$, the inversion process yields biologically valid protein or small molecule ligands that simultaneously achieve high affinity. We provide a detailed version of the inversion framework in Appendix A.

## 3 METHODOLOGY

We instantiate the four-stage framework introduced in Fig. 2 and propose MagicDock: **Stage1:** Docking-oriented ligand modeling, **Stage2:** Unsupervised pre-training, **Stage3:** Supervised fine-tuning, and **Stage4:** Inversion-based ligand generation. These modules constitute a de novo pipeline for receptor–ligand interactions. The pseudocode of our method is applied in Appendix G

### 3.1 STAGE 1: DOCKING-ORIENTED LIGAND MODELING

**Motivation.** We aim to develop a unified framework for modeling protein and molecule ligands, capturing shared structural principles while enabling effective modeling the docking process. Surface point clouds provide a compact, meaningful, SE(3)-equivariant representation, encoding geometric and chemical binding determinants, ideal for scalable pre-training and generative tasks.

**Surface Point Cloud Modeling.** We transform atomic structures into solvent-accessible surfaces, sampled as point clouds per (Wu & Li (2024)). The molecular surface is defined as the level set of a smooth distance function over atom centers, commonly referred to as the signed distance function (SDF). Candidate points $\{x_s^i\}$ are upsampled from Gaussian distributions around atomic coordinates $\{x_a^j\}$ and optimized via gradient descent using:

$$\text{SDF}(x_s^i) = -f(x_s^i) \cdot \log \sum_{j=1}^{N} \exp\left(-\frac{\|x_s^i - x_a^j\|}{\sigma_a^j}\right), \quad f(x_s^i) = \frac{\sum_{j=1}^{N} \exp(-\|x_s^i - x_a^j\|)\,\sigma_a^j}{\sum_{j=1}^{N} \exp(-\|x_s^i - x_a^j\|)}. \quad (4)$$

where $N$ is the number of atoms and $\sigma_a^j$ denotes the atomic radius. Multi-resolution point clouds are systematically generated with tailored scaling for protein ligands and molecule ligands, effectively balancing structural fidelity and computational performance.

**Feature Generation.** Each surface point is assigned highly comprehensive composite feature vectors seamlessly integrating chemical, atomic, and geometric characteristics:

$$f(x_i) = \text{concat}\big(f_{\text{chem}}(x_i),\ f_{\text{atom}}(x_i),\ f_{\text{geom}}(x_i)\big). \quad (5)$$

These features encode intrinsic and neighborhood information, tailored to the inherent characteristics of proteins and molecules, ensuring a unified, context-rich representation.

**Patch Partitioning.** To reduce computational complexity and facilitate Stage 2's pre-training, point clouds are partitioned into patches following (Wu & Li (2024)). Patch centers $X_c$ are obtained by farthest point sampling (FPS), and each center systematically groups its $K$ closest neighbors using $K$-nearest neighbor (KNN) search:

$$X_c = \text{FPS}(X_s), \quad X_c \in \mathbb{R}^{\rho M \times 3}, \qquad X_p = \text{KNN}(X_c, X_s), \quad X_p \in \mathbb{R}^{\rho M \times K \times 3}. \quad (6)$$

This patch-based structure effectively preserves local chemical–atomic-geometric context for docking and enables highly efficient unsupervised pre-training with discrete latent codes at the patch level. This optimized, docking-oriented representation seamlessly unifies receptors (i.e. protein) and ligands(i.e. protein and molecule), balancing biophysical accuracy and computational performance for subsequent modeling. We further demonstrate that MagicDock is approximately SE(3)-equivariant with respect to the initial position and orientation of the receptor in Appendix E.1. Comprehensive details of Stage 1 are provided in Appendix C.1.

## 3.2 STAGE 2: UNSUPERVISED PRE-TRAINING

**Motivation.** In Stage 2, we use the VQ-MAE framework (Wu & Li (2024)) on protein and molecule datasets, integrating mask autoencoding and quantization to learn transferable representations. Based on this, we have developed a pre-trained model that can generate high-quality representations of proteins and small molecules. (More details are applied in Appendix C.2 and C.3.)

**SE(3)-Equivariant Encoding.** The encoder systematically processes local surface patches $X_p$, with coordinates $\mathbf{x}_i \in \mathbb{R}^3$ and features $f_i$, using SE(3)-equivariant convolutions for significantly enhanced rigid-body consistency. The comprehensive resulting latent embedding for point $i$ is:

$$z_i = \sum_{j \in \mathcal{N}(i)} \sum_{l=0}^{L} R_l(\|\mathbf{x}_{ij}\|)\, Y_l\left(\frac{\mathbf{x}_{ij}}{\|\mathbf{x}_{ij}\|}\right) \cdot W_l f_j, \quad (7)$$

where $\mathbf{x}_{ij} = \mathbf{x}_j - \mathbf{x}_i$, $R_l(\cdot)$ are radial functions, $Y_l(\cdot)$ are spherical harmonics, and $W_l$ are weight matrices, guaranteeing equivariance under SE(3) spatial transformations.

**Masked Reconstruction with Vector Quantization.** Patches are masked at ratio $\delta = 50\%$, with masked and visible sets $X_{p,m} \in \mathbb{R}^{\delta \rho M \times K \times 3}$ and $X_{p,vis} \in \mathbb{R}^{(1-\delta)\rho M \times K \times 3}$. Masked patch tokens are quantized using a learnable codebook via Gumbel-Softmax relaxation. Visible and quantized embeddings are systematically decoded to reconstruct: (i) spatial coordinates via

$$\hat{X} = \text{Reshape}(\text{MLP}(H_p^{(L_2)})), \quad \hat{X} \in \mathbb{R}^{\delta \rho M \times K \times 3}, \quad (8)$$

and (ii) surface curvature derived from the covariance matrix of each carefully masked patch, with pseudo-curvatures $\psi_i = \epsilon_i / \sum_{j=1}^{3} \epsilon_j$ accurately predicted by an MLP.

**Training Objective.** The loss effectively combines accurate coordinate reconstruction, precise curvature prediction, and comprehensive KL divergence regularization:

$$\mathcal{L} = \nu_1 \mathcal{L}_{\text{rec}}(X_{p,m}, \hat{X}) + \nu_2 \mathcal{L}_{\text{cur}}(\psi, \hat{\psi}) + \nu_3 \mathcal{L}_{\text{KL}}(q(Z_{p,m} \mid H_{p,m}), p(Z_{p,m})), \quad (9)$$

where $\mathcal{L}_{\text{rec}}$ uses Chamfer distance, $\mathcal{L}_{\text{cur}}$ uses RMSE, and $H_{p,m}$ denotes the masked patch embeddings before quantization. This design effectively embeds rich ligand surface semantics, ensuring robust reconstruction fidelity and enhanced geometric consistency. More details are in Appendix C.2.

### 3.3 STAGE 3: SUPERVISED FINE-TUNING

**Motivation.** The models obtained in Stage 2 provide high-quality representations of protein and small molecule data, but they cannot be directly used for docking tasks. Therefore, in Stage 3, we use SE(3)-equivariant attention to capture receptor-ligand dependencies and fine-tune the model using ground-truth labeling on three progressively downstream tasks: Pocket Prediction, Interaction Prediction, and Binding-Affinity Regression.

**Equivariant Attention Fusion.** Receptor and ligand point clouds are independently encoded by Stage 2's encoders into latent fields $Z_r \in \mathbb{R}^{N_r \times d}$ and $Z_l \in \mathbb{R}^{N_l \times d}$. Interfacial dependencies are captured through an SE(3)-equivariant attention followed by a permutation-invariant aggregator:

$$\text{Attn}(Z_r, Z_l) = \text{softmax}\left(\frac{Q_r^{(\ell=0)}(K_l^{(\ell=0)})^\top}{\sqrt{d}}\right) V_l, \quad \tilde{z} = \mathcal{A}\big(Z_r, \text{Attn}(Z_r, Z_l)\big) \in \mathbb{R}^{2d}, \quad (10)$$

where $Q_r^{(\ell=0)} = Z_r^{(\ell=0)} W_Q$, $K_l^{(\ell=0)} = Z_l^{(\ell=0)} W_K$, and $V_l = Z_l W_V$ are computed with learnable matrices $W_Q, W_K, W_V \in \mathbb{R}^{d \times d}$. The attention scores are derived from $\ell = 0$ (scalar) channels to ensure SE(3)-invariance, while higher-order features in $V_l$ are transformed via Wigner-$D$ matrices.

**Multi-Task Supervision.** Three progressively objectives align representations with docking semantics: (i) Pocket segmentation classifies receptor embeddings $z_i \in Z_r$ with a loss combining binary cross-entropy (BCE) and a geometric regularization term; (ii) Interaction prediction uses the fused representation $\tilde{z}$ with BCE loss; (iii) Binding affinity regression predicts binding free energy via mean squared error (MSE). The corresponding objectives are summarized as:

$$\mathcal{L}_{\text{pocket}} = \frac{1}{N_r} \sum_{i=1}^{N_r} \text{BCE}(\hat{y}_i, y_i) + \lambda_p \mathcal{R}_{\text{geom}}, \quad \mathcal{L}_{\Delta G} = \frac{1}{|\mathcal{V}|} \sum_{(r,l) \in \mathcal{V}} \big(\hat{y}_{\Delta G}(r, l) - \Delta G(r, l)\big)^2. \quad (11)$$

The overall fine-tuning objective is:

$$\mathcal{L}_{\text{FT}} = \alpha \mathcal{L}_{\text{pocket}} + \beta \mathcal{L}_{\text{int}} + \mathcal{L}_{\Delta G}, \quad (12)$$

where $\mathcal{L}_{\text{int}}$ is the BCE loss for interaction prediction, and $\alpha, \beta, \lambda_p > 0$ are tuned on validation data. Task-specific MLPs, the equivariant attention module, and encoders are optimized jointly.

### 3.4 STAGE 4: INVERSION-BASED LIGAND GENERATION

**Motivation.** To convert the docking-aware backbone into a generative model, we use an inversion framework that optimizes ligands' structure and feature in a continuous surface embedding space. In addition, this flexible inversion-based pipeline can effectively utilize the unified differentiable data structure to generate different categories of ligands in one pipeline.

**Gradient-driven Inversion.** For a receptor $R$ and initial ligand $S_0$, Stage 3 encoders systematically produce latent fields $Z_r$ and $Z_l$, fused via SE(3)-equivariant attention into $\tilde{z}$ (Eq. 10). The composite objective $\mathcal{F}(S; R, \Theta)$ (Eq. 43) effectively reflects pocket consistency, interaction plausibility, and binding affinity. Ligand point-cloud coordinates $\mathbf{x}$ and features $f$ are rigorously optimized by descending $\nabla_{(\mathbf{x}, f)} \mathcal{F}$, with updates mapped to chemically valid structures via:

$$S^\star = \lim_{t \to \infty} \mathcal{G}_{\text{type}}\Big(\Pi_{\mathcal{C}_{\text{valid}}}\big((\mathbf{x}, f)^t - \eta_t \nabla_{(\mathbf{x}, f)} \mathcal{F}(S_t; R, \Theta)\big)\Big), \quad (13)$$

where $\eta_t$ is the step size, $\Pi_{\mathcal{C}_{\text{valid}}}$ ensures chemical and geometric validity, $\mathcal{G}_{\text{type}}$ decodes point clouds into atomistic graphs, and $S_t$ denotes the ligand structure at iteration $t$. This unifies representation learning and structure generation for docking-aware ligands. Details are in Appendix C.4.

## 4 THEORETICAL ANALYSIS

To provide a cohesive theoretical foundation, we present MagicDock's inversion-based framework, grounded in rigorous analyses in Appendix E. Theorem 1 establishes SE(3)-equivariance across all stages, ensuring rotational and translational invariance in ligand generation. Theorem 2 proves convergence of the projected gradient descent in the inversion phase to stationary points under smoothness and boundedness assumptions. Theorem 3 demonstrates theoretical superiority over decoupled

Table 1: Performance comparison on the SKEMPI v2 (left) and SAbDab (right). Best results are in **bold**, second best are underlined. Arrows indicate whether higher (↑) or lower (↓) values are better.

| Method | IMP (↑) | STA (↑) | DIV (↑) | NOV (↑) | AAR (↑) |
|--------|---------|---------|---------|---------|---------|
| RAbD | 21.58/15.12 | 0.779/0.763 | 0.728/0.764 | 0.838/0.829 | 38.23/38.75 |
| DiffAB | 17.65/11.69 | 0.703/0.736 | 0.744/0.754 | 0.820/0.868 | 41.65/38.52 |
| HSRN | 23.19/17.72 | 0.844/0.809 | 0.801/0.805 | 0.877/0.924 | 45.29/40.63 |
| dyMEAN | 15.93/8.55 | 0.821/0.799 | 0.805/0.799 | 0.923/0.953 | 44.67/44.06 |
| ABDPO | 25.17/16.63 | 0.853/0.833 | 0.796/0.816 | 0.905/0.932 | 46.19/43.53 |
| Abx | 28.76/21.80 | 0.866/0.820 | 0.812/0.820 | 0.889/0.950 | 46.35/44.29 |
| Ours | **36.32/27.87** | **0.874/0.851** | **0.815/0.824** | **0.934/0.957** | **48.73/46.14** |

Table 2: Performance comparison on the PDBBind2020 (left) and CrossDocked2020 (right).

| Method | Vina (↓) | Affinity (↑) | STA (↑) | DIV (↑) | NOV (↑) | QED (↑) |
|--------|----------|--------------|---------|---------|---------|---------|
| DockStream | -5.51/ -5.15 | 15.13/17.86 | 0.765/0.780 | 0.717/0.621 | 0.985/0.967 | 0.455/0.401 |
| 3D-SBDD | -6.35/ -6.24 | 27.88/28.54 | 0.853/0.801 | 0.768/0.701 | 0.997/0.998 | 0.503/0.483 |
| liGAN | -6.03/ -6.11 | 22.56/22.15 | 0.830/0.825 | 0.772/0.663 | 0.997/0.997 | 0.489/0.377 |
| ALIDIFF | -7.21/ -6.79 | 35.74/**65.63** | **0.875**/0.833 | 0.756/0.727 | 0.998/0.999 | 0.472/0.464 |
| DRUGFLOW | -7.12/-6.81 | 33.56/52.35 | 0.858/0.838 | 0.764/0.718 | 0.995/0.999 | 0.520/0.519 |
| DIFFSBDD | -6.99/-6.86 | 28.83/35.88 | 0.867/0.825 | **0.801**/0.705 | **1.000**/0.998 | 0.511/0.502 |
| Ours | **-7.36/-7.02** | **40.63**/60.02 | 0.866/**0.840** | 0.778/**0.730** | **1.000/1.000** | **0.552/0.544** |

generate-and-optimize paradigms, with reachable objective sets strictly contained and improved under generator misspecification. Theorems 4, 5 highlight efficiency advantages, outperforming traditional methods in computational cost. Theorem 6 shows information-theoretic optimality via maximized mutual information $I(X;Y)$, balancing high output entropy and low conditional uncertainty. Finally, Theorem 7 underscores lower sample complexity, $O(\log 1/\epsilon)$ for fine-tuning, leveraging pre-training for data efficiency over baselines' $O(1/\epsilon)$.

## 5 EXPERIMENTS

To validate the distinct superiority of MagicDock, we conduct a series of rigorous and comprehensive experiments on diverse datasets for both molecule and protein ligand. We aim to answer: **Q1** (Effectiveness): Does MagicDock outperform state-of-the-art baselines in generating ligand? **Q2** (Interpretability): What enables MagicDock to effectively produce high-affinity ligands? **Q3** (Robustness): Is MagicDock resilient to structural noise, biological variability, and does it exhibit reliable convergence? **Q4** (Efficiency): Does MagicDock achieve superior trade-offs in runtime, resource usage, scalability, and data efficiency during ligand generation?

### 5.1 OVERALL PERFORMANCE (Q1)

To answer **Q1**, we conducted a systematic evaluation of the effectiveness of ligand design for two scenarios. In the following, we list the detailed experimental settings for these two scenarios.

**Datasets & Baselines & Evaluation Metrics.** We utilize tailored datasets, baselines, and evaluation metrics for both protein and small molecule ligand design, with details provided in Appendix D. Furthermore, we have made fair adjustments for all baseline methods to adapt to challenging de novo ligand design scenario, as comprehensively detailed in the Appendix H.

**Experimental Results.** As shown in Table 1 and Table 2, MagicDock outperforms all baselines on both benchmarks, achieving better binding affinity, stability, diversity and novelty, with superior AAR for proteins and higher QED for molecules. MagicDock's experimental affinity performance is slightly lower than AliDiff's due to the latter's direct exact energy alignment via preference optimization on pre-trained diffusion models, which efficiently biases toward high-affinity sample distributions but sacrifices generation diversity and incurs substantial computational overhead. Hyperparameter details are applied in Appendix L.

## 5.2 INTERPRETABILITY STUDY (Q2)

Having established the remarkable effectiveness in Q1, we now turn to Q2 and investigate why MagicDock can produce high-affinity ligands. Specifically, we first analyzed the contribution of each stage to the final result and whether the gradient of the inversion target provided meaningful biological localization signals. In addition, we also visualized the process of ligand generation.

**Ablation Study.** To disentangle the role of each stage, we conduct module-wise ablations: (i) *w/o Stage 1*: replacing surface point clouds with raw atom graphs; (ii) *w/o Stage 2*: removing unsupervised pre-training; (iii) *w/o Stage 3*: disabling supervised fine-tuning; (iv) *w/o Stage 4*: substituting gradient-guided inversion with exhaustive element-type search. Fig. 6a, 6b show that each stage plays a critical role, with the full pipeline achieving the best performance.

**Gradient Attribution & Localization.** We further ask whether gradients from the composite inversion objective $\mathcal{F}(S; R, \Theta)$ (Appendix C.4) effectively localize on biologically meaningful binding sites learned in Stage 3. We compute Integrated Gradients on receptor and ligand surfaces and systematically evaluate whether high-attribution regions align with ground-truth interfaces. Fig. 3 demonstrates that inversion not only optimizes affinity but also consistently highlights mechanistically significant relevant regions, addressing Q2.

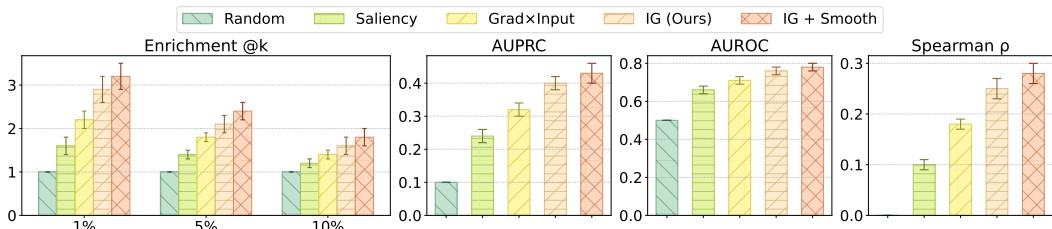

Figure 3: Performance comparison of attribution methods for interpretability, evaluated on Enrichment @k (1%, 5%, 10%), AUPRC, AUROC, and Spearman correlation.

**Local Perturbation Consistency.** We assess whether gradient attributions predict energetic effects of local edits by introducing small perturbations near high-attribution sites and comparing predicted with actual $\Delta\mathcal{F}$. This tests whether attributions enable actionable refinement. As shown in Fig. 8, our method outperforms saliency and Grad×Input, approaching physics-based Rosetta evaluations.

**Case Study.** To evaluate the significant impact of constraints in MagicDock's inversion-based ligand generation process, we generated two ligands for the target receptor. At each iteration, we systematically assessed binding affinity, as shown in Fig. 4 and Fig. 12. Additionally, we have listed a series of molecular ligands generated by MagicDock, as shown in Fig. 11 and Fig. 13 in Appendix K.

## 5.3 ROBUSTNESS STUDY (Q3)

Having established remarkable interpretability, we next address Q3: whether MagicDock is consistently robust to input variations and biological uncertainty. We design three sets of experiments systematically probing artificial noise, realistic receptor perturbations, and convergence properties.

**Geometric and Feature Noise.** We assess robustness on 100 receptor–ligand pairs by adding Gaussian coordinate noise and feature dropout to receptor surfaces, then comparing ligand generation on noisy versus clean inputs (Fig. 7). Results show that MagicDock consistently outperforms baselines under perturbations, demonstrating stable performance and robustness to biological uncertainty.

**Conformational and Mutational Variability.** We further evaluate remarkable robustness to biological variability by subjecting 100 receptors to perturbations, including conformer ensembles, single-point mutations, and combined variants. These perturbed structures are systematically processed through the inversion pipeline. Performance degradation is assessed via IMP for protein baselines and High-Affinity for molecule baselines. As shown in Fig. 9, statistical tests compare conservative and non-conservative mutations, illustrating MagicDock's exceptional robustness.

**Convergence Study.** The convergence of MagicDock's gradient-driven inversion stage is rigorously established in Appendix E.2. The proof systematically demonstrates that, under standard assump-

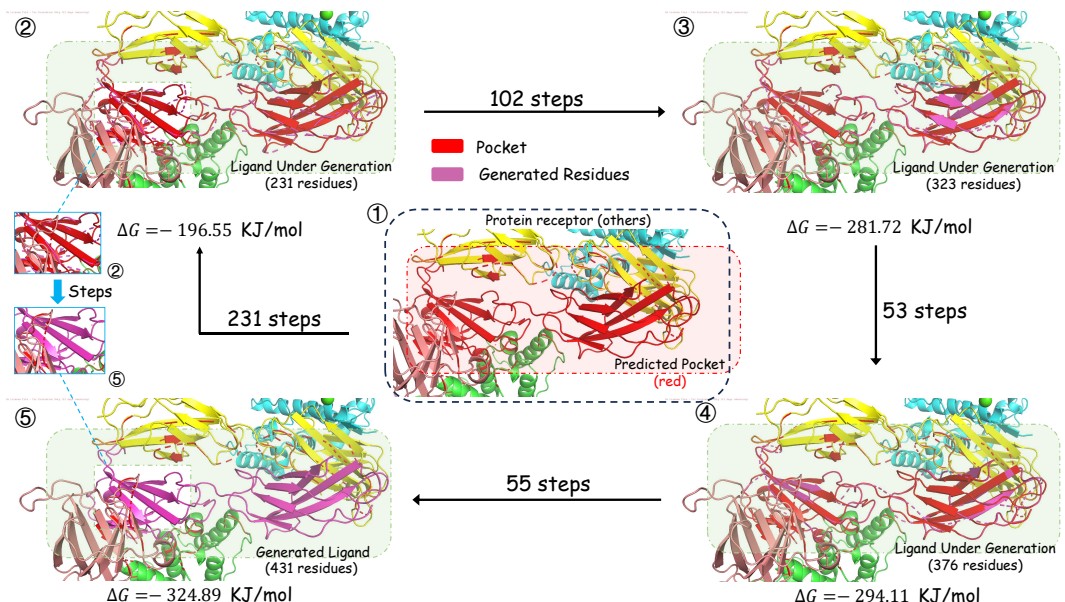

Figure 4: Visualization of an example of generated protein-ligand complexes.

tions for projected gradient descent, the ligand sequence $P_{\text{lig}}^t$ converges to a stationary point of the composite objective $\mathcal{F}$, with $\min_t |\nabla \mathcal{F}(P_{\text{lig}}^t)|^2 \to 0$ as $t \to \infty$. This guarantees that MagicDock consistently optimizes ligands with exceptionally high reliability.

### 5.4 EFFICIENCY STUDY (Q4)

Finally, we address Q4 by systematically quantifying whether MagicDock achieves a highly favorable trade-off between computational cost and accuracy. Having shown that the framework is both interpretable and robust, we now analyze runtime efficiency and comprehensive scalability.

**Runtime and Resource Usage.** We benchmark MagicDock against baselines using 100 receptors, generating one ligand per receptor under identical hardware. Wall-clock time and memory usage are assessed per computational stage. Fig. 10a demonstrates MagicDock's reduced runtime while Fig. 10b highlights lower peak memory consumption, underscoring its resource efficiency.

**Scalability.** Scalability is evaluated across nine settings varying receptor sizes and ligand complexities using 100 receptors from CrossDocked2020 and SAbDab. Metrics include per-iteration latency, iterations-to-converge, memory footprint, and scaling exponent $\gamma$ from $T \propto N^\gamma$. Table 5 and Table 6 shows MagicDock's sub-quadratic scaling ($\gamma = 1.4$), enabling real-time deployment.

**Data Efficiency.** As established in Appendix E.6, MagicDock's inversion framework achieves $\epsilon$-accuracy in supervised fine-tuning with only $O(1/\epsilon)$ samples (up to logarithmic confidence factors), compared to $O(1/\epsilon^2)$ for GANs and $O(T/\epsilon^2)$ for diffusion models. This linear-in-$1/\epsilon$ sample complexity, enabled by pre-training's strong convexity and low effective dimension, effectively ensures robust generalization in challenging data-scarce docking tasks with limited annotated complexes.

### 6 CONCLUSION

In this study, we introduced MagicDock, an inversion-based framework unifying generation and optimization in a streamlined, docking-driven workflow for de novo ligand design. Using surface point-cloud modeling, SE(3)-equivariant pretraining, and docking-oriented fine-tuning, MagicDock eliminates external priors for robust de novo design of protein and small-molecule ligands. Experiments show strong effectiveness and efficiency. These results position inversion-based docking as a versatile paradigm overcoming traditional limitations for practical ligand design. Limitations and future work are applied in Appendix F.

## REPRODUCIBILITY STATEMENT

To ensure reproducibility, we provide detailed MagicDock architecture descriptions, theoretical analysis, and objectives in Sec. 3 and Sec. 5 and Appendix A to Appendix D. Datasets for pre-training and evaluation, including processing, splits, and sources, are in Appendix E. Hyperparameters, training, and ablations are in Appendix L. Codes and other necessary materials are provided in our supplementary materials.

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

# APPENDIX

## TABLE OF CONTENT

# A  THE INVERSION FRAMEWORK IN DETAILS

## A.1  MODEL TRAINING

Recent advances in generative modeling have increasingly embraced the Inversion framework as a powerful approach, which reverses traditional data flow by reconstructing or optimizing structures from latent representations or gradients, as demonstrated in InversionGNN( Niu et al. (2025)). This framework is particularly well-suited for our docking-optimized design of proteins and small molecules, offering a flexible paradigm for zero-start generation. Let us guide you through its structure step by step, focusing on the unsupervised pre-training and supervised fine-tuning phases that establish a robust foundation for subsequent inversion-based generation.

① Unsupervised Pre-Training. This phase lays the foundation by initializing the Inversion framework through an unsupervised training process that encodes input data into a latent representation, capturing transferable geometric and chemical priors without relying on labeled data. Generally, it involves an encoder that transforms structured input data—comprising $n$ elements and their relationships—into hidden embeddings $\mathbf{z} \in \mathbb{R}^d$, updated iteratively using a propagation rule. The general update can be expressed as:

$$\mathbf{z}_{t+1} = \text{Prop}\left(\mathbf{z}_t, \{\mathbf{z}_v \mid v \in \mathcal{N}(u)\}, \theta\right), \tag{14}$$

where $\theta$ represents trainable parameters, and Prop denotes a propagation function tailored to the data structure, which may include graphs, point clouds, or other relational formats. To ensure adaptability, the latent space is refined with feedback from an objective function, requiring differentiability for gradient-based optimization.

In our work, this phase employs a pre-trained SE(3)-equivariant encoder to process the interaction graph derived from 3D point clouds, generating embeddings $\mathbf{h}_u \in \mathbb{R}^d$ as:

$$\mathbf{h}_u^{(l)} = \text{Prop}_{\text{SE(3)}}\left(\mathbf{h}_u^{(l-1)}, \left\{\mathbf{h}_v^{(l-1)} \mid v \in \mathcal{N}(u)\right\}\right), \tag{15}$$

where the SE(3)-equivariant propagation weights are optimized using self-supervised objectives, such as masked reconstruction and vector quantization, to embed rich surface semantics. This establishes a robust, docking-aware representational backbone by learning from extensive datasets of proteins and small molecules, ensuring geometric consistency and chemical plausibility without explicit supervision.

② Supervised Fine-Tuning. Building on the unsupervised pre-training, this phase refines the model with task-specific supervision to align representations with docking semantics, incorporating ground-truth labels from protein–ligand complexes. The fine-tuning calibrates the pre-trained encoder to capture interfacial dependencies and binding signals, using objectives like pocket segmentation, interaction prediction, and binding affinity regression.

The receptor and ligand point clouds are encoded into latent fields $\mathbf{Z}_r$ and $\mathbf{Z}_l$, fused via cross-attention to model interactions. The multi-task loss integrates these objectives, with gradients from docking energy $E_{\text{dock}}$, defined as $\nabla_{\mathbf{h}_u} E_{\text{dock}} = \frac{\partial E_{\text{dock}}}{\partial \mathbf{h}_u}$, providing feedback to fine-tune the parameters. This supervised refinement enhances the model's ability to predict high-affinity structures, mitigating issues like representation collapse and enabling seamless transition to the inversion phase for generation.

The trainable components across both phases include the encoder weights and gradient optimization parameters, collectively parameterized by $f_\theta$. This two-phase approach—unsupervised pre-training followed by supervised fine-tuning—mitigates issues like model degradation (from suboptimal initialization) and gradient misalignment, with optimization dynamics driven by gradient-based supervision for improved convergence in docking-oriented ligand design.

## A.2  INVERSION PHASE

Building on the trained model, this phase reconstructs the desired structures by reversing the encoding process through gradient-driven optimization. Generally, the inversion process updates a latent or structural representation $\mathbf{x}$ based on an objective function $L$, formulated as:

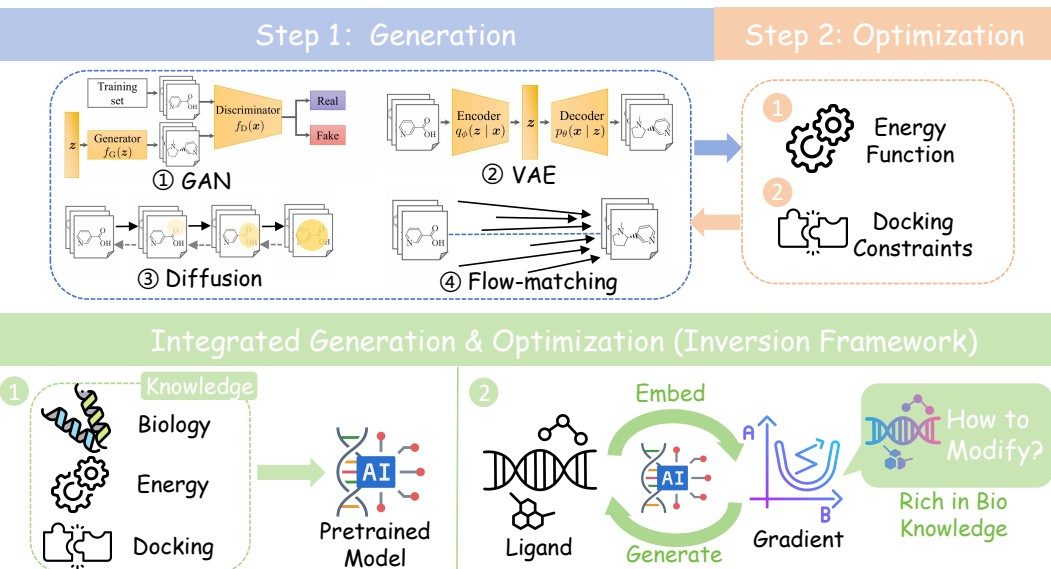

Figure 5: Comparison of the inversion framework with other generative ligand design methods.

$$\mathbf{x}_{t+1} = \mathbf{x}_t - \eta \nabla_{\mathbf{x}_t} L(\mathbf{x}_t), \tag{16}$$

where $\eta$ is the learning rate, and $L$ incorporates domain-specific constraints. In our implementation, the Inversion generation module refines point cloud positions $\mathbf{p}_u$ and features $\mathbf{f}_u$ using gradient adjustments guided by docking feedback, expressed as:

$$\mathbf{x}_{t+1} = \text{InvGen}\left(\mathbf{p}_u + \alpha \nabla_{\mathbf{p}_u} E_{\text{dock}}, \mathbf{f}_u + \beta \nabla_{\mathbf{f}_u} E_{\text{dock}}, C_{\text{bio}}\right), \tag{17}$$

where $\alpha$ and $\beta$ are learning rates, and $C_{\text{bio}}$ enforces biochemical constraints. This enables reverse inference of protein sequences or small molecule structures from zero-start, simultaneously optimizing both molecular types.

The trainable components of this framework include the encoder weights and gradient optimization parameters, collectively parameterized by $f_\theta$. This two-phase approach mitigates issues like model degradation (from suboptimal initialization) and representation collapse (from gradient misalignment), with optimization dynamics driven by gradient-based supervision for improved convergence.

## B  RELATED WORKS IN DETAILS

Following the taxonomy in the main text, we reorganize existing approaches into four paradigms according to how generation is coupled with optimization. For completeness, we further discuss them from two complementary perspectives: protein design and molecular ligand design.

① **Decoupled Paradigms.** (a) *Protein design.* Classical protein–protein docking pipelines, such as ZDOCK Chen et al. (2010), HADDOCK Dominguez et al. (2003), and RosettaDock Lyskov & Gray (2008), employ geometric complementarity and energy minimization, sometimes with partial priors like known paratopes Kozakov et al. (2017); Yan et al. (2020); Ganea et al. (2021). Deep learning extensions such as AlphaFold-Multimer Evans et al. (2022) improve accuracy but remain resource-intensive. Early generative efforts, e.g., RNN-based paratope generation Liu et al. (2020); Saka et al. (2021) or sequence–structure joint models Jin et al. (2021), also separated sampling from docking-based optimization.

(b) *Molecular ligand design.* Structure-based drug discovery relies on docking engines including AutoDock (Morris et al. (2008)), DOCK (Ewing et al. (2001)), Glide (Halgren et al. (2004)), and GOLD (Verdonk et al. (2003)), supported by scoring functions and refinement via MD, FEP, or

MM/PBSA simulations (Berdigaliyev & Aljofan (2020); Fabricant & Farnsworth (2001)). These workflows represent ligands in predefined descriptors, then optimize poses post hoc. Early generative models (VAEs (Gómez-Bombarelli et al. (2018); Skalic et al. (2019)), GANs (Guimaraes et al. (2017)) also adopted a decoupled generation–evaluation scheme. While interpretable, such methods are loosely tied to binding or multi-objective constraints.

② **Implicitly Coupled Paradigms.** (a) *Protein ligand design.* Here, optimization signals are embedded into stochastic generative processes. Graph-based generative methods (Jin et al. (2021)), backbone-conditioned models (Ingraham et al. (2019)), and energy-aware strategies (Tischer et al. (2020); Cao et al. (2021)) guide sequence/structure generation with implicit docking feedback. Diffusion-style models (e.g., DiffAb, HSRN) refine CDR loops and interfaces with SE(3)-equivariant priors.

(b) *Molecule ligand design.* Diffusion-based models such as DiffDock (Corso et al. (2022)) and GeoDiff (Xu et al. (2022)) integrate $\Delta G$ or docking signals during denoising. Pocket2Mol (Peng et al. (2022)) and TankBind (Lu et al. (2022)) condition on receptor pockets. Broader frameworks, including DockStream (Guo et al. (2021)), ALIDIFF (Gu et al. (2024)), and DRUGFLOW (Schneuing et al. (2025)), couple docking or ADMET constraints with molecular diffusion or generative flows (Shi et al. (2020); Lee et al. (2023)). VAE (Liu et al. (2018); Fu et al. (2020); Griffiths & Hernández-Lobato (2020); Wang et al. (2022)) and GAN-based methods (De Cao & Kipf (2018); Abbasi et al. (2022)) also benefit from structural conditioning. Despite progress, efficiency and smooth latent space learning (Brown et al. (2019); Huang et al. (2021); Gao et al. (2024b)) remain challenges.

③ **Surrogate- or Heuristic-coupled Paradigms.** (a) *Protein ligand design.* Reinforcement learning has been applied to antibody optimization (Ingraham et al. (2019); Tischer et al. (2020)), while memory-augmented models integrate docking oracles into generation. Energy-based approaches similarly exploit docking rewards (Cao et al. (2021)), yet suffer from high computational cost.

(b) *Molecule ligand design.* Ligand optimization often leverages search in discrete chemical space. Reinforcement learning (Ståhl et al. (2019); You et al. (2018); Zhou et al. (2019); Gottipati et al. (2020); Gao et al. (2024a); Jain et al. (2023)), evolutionary algorithms (Jensen (2019); Nigam et al. (2019); Chen et al. (2021)), Markov Chain Monte Carlo (Xie et al. (2021); Fu et al. (2021b)), tree search (Ma et al. (2021)), and Bayesian optimization (Korovina et al. (2020); Moss et al. (2020)) iteratively refine molecules with docking oracles. While effective, they require many evaluations and struggle with balancing multiple objectives (Blum & Roli (2003); Mazyavkina et al. (2021)). Frameworks like DockStream (Guo et al. (2021)), ALIDIFF (Gu et al. (2024)), and DRUGFLOW (Schneuing et al. (2025)) attempt to reduce cost via chemical and geometric priors.

④ **Explicitly Gradient-coupled Paradigm (Ours).**

Inversion-based frameworks directly couple generation and optimization via gradient guidance in latent space. Unlike stochastic or heuristic strategies, gradients provide more efficient and interpretable refinements of sequence–structure pairs (Niu et al. (2025); Fu et al. (2021a)). The proposed **MagicDock** exemplifies this paradigm by unifying 3D geometry, docking constraints, and biophysical objectives. By treating ligand generation as a gradient-driven process, it co-optimizes $\Delta G$, specificity, and stability without handcrafted schedules. This enables biologically relevant designs under a single, generalizable architecture.

## C  METHODOLOGY IN DETAILS

### C.1  STAGE 1: DOCKING-ORIENTED LIGAND MODELING

We represent ligands (both proteins and molecules) through solvent-accessible surfaces parameterized by a probe radius $r_{\text{probe}} = 1.4\,\text{Å}$. Let $C = \{(c_j, r_j)\}_{j=1}^{M}$ denote atomic centers and van der Waals radii. The molecular surface is defined as the iso-level set of the distance field:

$$S = \{x \in \mathbb{R}^3 \mid d(x; C) = r_{\text{probe}}\}, \quad d(x; C) = \min_j \|x - c_j\| - r_j, \tag{18}$$

where the minus sign ensures consistency with van der Waals boundaries. A differentiable representation is achieved using a smoothed distance function:

$$\text{SDF}(x) = -\bar{\sigma}(x) \log \sum_{j=1}^{M} \exp\left(-\frac{\|x - c_j\|}{\sigma_j}\right), \quad \bar{\sigma}(x) = \frac{\sum_j \exp(-\|x - c_j\|)\sigma_j}{\sum_j \exp(-\|x - c_j\|)}. \tag{19}$$

Candidate points, sampled from Gaussian perturbations around atoms, are iteratively projected onto the iso-surface via gradient descent. A triangulated mesh is constructed using the ball-pivoting algorithm (BPA) with radii in $[r_{\text{probe}}, 2r_{\text{probe}}]$, and $N = 5000$ points are uniformly sampled for proteins (scaled for ligands). Surface normals are estimated by weighted PCA within a $2r_{\text{probe}}$ neighborhood:

$$\Sigma_i = \sum_{j \in \mathcal{N}_n(i)} w_{ij}(x_j - \bar{x}_i)(x_j - \bar{x}_i)^\top, \quad w_{ij} = \exp\left(-\frac{\|x_i - x_j\|^2}{(2r_{\text{probe}})^2}\right), \tag{20}$$

where $\mathcal{N}_n(i)$ is the set of neighboring points for normal estimation.

Each surface point $x_i$ is enriched with a feature vector integrating chemical descriptors, atomic type indicators, geometric descriptors, coordinates, and a molecule-type identifier $I_i$:

$$f(x_i) = \text{concat}\big(f_{\text{chem}}(x_i), f_{\text{atom}}(x_i), f_{\text{geom}}(x_i), I_i, x_i\big). \tag{21}$$

**Atomic features.** Atomic features $f_{\text{atom}}(x_i)$ encode the local element distribution through weighted one-hot statistics. For proteins, we adopt $\{C, H, O, N, S, Se\}$ (6D). For small molecules, we adopt $\{C(sp3), C(sp2), H, O(sp3), O(sp2), N(sp3), N(sp2), S(sp2)\}$ (8D). The statistics are aggregated using probe-scaled weights $\omega_{ia} = 1/(\|x_i - c_a\|/r_{\text{probe}} + \varepsilon)$. *Note that this atomic vocabulary is limited to common bio-organic atoms, while halogens and metal ions frequently appearing in pharmaceutically relevant compounds are not yet included; we discuss extending the feature space in Section F.*

**Chemical features.** Chemical descriptors $f_{\text{chem}}(x_i)$ are continuous values that capture local physicochemical properties from neighboring atoms within $r_{\text{chem}} = 5.0\,\text{Å}$. Specifically, we define: hydrogen-bonding potential,

$$H_{\text{bond}}(x_i) = \frac{\sum_{a \in \mathcal{N}_c(i)} \omega_{ia} \mathbb{1}[t_a \in \{O, N, S\}]}{\sum_{a \in \mathcal{N}_c(i)} \omega_{ia}}, \tag{22}$$

charge polarity,

$$C(x_i) = \frac{\sum_{a \in \mathcal{N}_c(i)} \omega_{ia} \left(\mathbb{1}[t_a \in \{O, N\}] - \mathbb{1}[t_a \in \{C, S\}]\right)}{\sum_{a \in \mathcal{N}_c(i)} \omega_{ia}}, \tag{23}$$

and hydrophobicity/aromaticity,

$$H_{\text{phob}}(x_i) = \frac{\sum_{a \in \mathcal{N}_c(i)} \omega_{ia} \mathbb{1}[t_a = C]}{\sum_{a \in \mathcal{N}_c(i)} \omega_{ia}}, \quad A_{\text{aro}}(x_i) = \text{clip}\left(\frac{H_{\text{phob}}(x_i)}{4.5}, 0, 1\right). \tag{24}$$

Here, $\mathcal{N}_c(i)$ denotes the chemical neighborhood, $\mathbb{1}[\cdot]$ the indicator function, and $\text{clip}(x, a, b)$ truncates $x$ into $[a, b]$.

**Geometry features.** Atomic statistics $T_{i,k}$ reflect weighted element distributions, with 6D for proteins and 8D for small molecules. Geometric descriptors from a $k_g = 10$ nearest-neighbor set $\mathcal{N}_g(i)$ include mean curvature $\kappa_{i,1}$, Gaussian curvature $\kappa_{i,2}$, and local density $D_i$, computed as:

$$\kappa_{i,1} = \frac{1}{|\mathcal{N}_g(i)|} \sum_{j \in \mathcal{N}_g(i)} \|n_i - n_j\|, \tag{25}$$

$$\kappa_{i,2} = \prod_{m=1}^{3} \epsilon_m, \quad \{\epsilon_1, \epsilon_2, \epsilon_3\} = \text{eig}\left(\frac{1}{|\mathcal{N}_g(i)|} \sum_{j \in \mathcal{N}_g(i)} (x_j - \bar{x}_i)(x_j - \bar{x}_i)^\top\right), \tag{26}$$

$$D_i = \frac{1}{k_g} \sum_{j \in \mathcal{N}_g(i)} \|x_i - x_j\|, \quad \text{normalized to } [0, 1], \tag{27}$$

where $\text{eig}(\cdot)$ returns eigenvalues of the covariance matrix.

To enhance scalability, point clouds are partitioned into overlapping patches. Centers are obtained via farthest point sampling (FPS), each grouping $K = 50$ neighbors within $r_{\text{patch}} = 5.0\,\text{Å}$:

$$\mathcal{P}(x_c) = \{x_j \mid x_j \in \text{KNN}(x_c, K), \|x_j - x_c\| \leq r_{\text{patch}}\}, \tag{28}$$

where KNN denotes the $K$-nearest neighbors of a center point. Patch features are summarized by mean and variance pooling:

$$\mu_{k,f} = \tfrac{1}{|\mathcal{P}_k|} \sum_{x_i \in \mathcal{P}_k} f(x_i), \quad \sigma^2_{k,f} = \tfrac{1}{|\mathcal{P}_k|} \sum_{x_i \in \mathcal{P}_k} (f(x_i) - \mu_{k,f})^2, \tag{29}$$

with interface labels assigned for points within 4.0 Å of ligand or 2.0 Å of protein surfaces.

This yields a probe-aware, feature-enriched, patch-structured representation, unifying proteins and ligands for pre-training and generative modeling.

## C.2 STAGE 2: UNSUPERVISED PRE-TRAINING

In the second stage, the encoder is pre-trained in an unsupervised manner to capture transferable docking-aware priors from proteins and small molecules. The framework integrates SE(3)-equivariant convolutions, patch-level masking with vector quantization, and reconstruction objectives targeting geometric and physicochemical properties, establishing a robust foundation for subsequent fine-tuning. Below, we detail the computational framework and objectives, ensuring seamless integration of encoding and reconstruction processes.

**SE(3)-Equivariant Encoding.** Each molecular surface is represented as a collection of patches

$$X_p = \{(\mathbf{x}_i, f_i)\}_{i=1}^K,$$

where $\mathbf{x}_i \in \mathbb{R}^3$ are Cartesian coordinates, $f_i \in \mathbb{R}^d$ are feature vectors of dimension $d$, and $K$ is the total number of patches. The encoder aggregates local neighborhoods $\mathcal{N}(i)$ using SE(3)-equivariant convolutions:

$$z_i = \sum_{j \in \mathcal{N}(i)} \sum_{l=0}^{L} R_l(\|\mathbf{x}_{ij}\|) \, Y_l\left(\tfrac{\mathbf{x}_{ij}}{\|\mathbf{x}_{ij}\|}\right) \cdot W_l f_j, \tag{30}$$

where $\mathbf{x}_{ij} = \mathbf{x}_j - \mathbf{x}_i$, $R_l(\cdot)$ are learnable radial functions, $Y_l(\cdot)$ are spherical harmonics of order $l$, $W_l$ are trainable matrices, and $L$ is the maximum order of harmonics. This ensures equivariance under rigid motions $(R, t)$ with $R \in SO(3)$ and $t \in \mathbb{R}^3$:

$$z_i(R\mathbf{x} + t) = \rho(R) \, z_i(\mathbf{x}), \tag{31}$$

where $\rho(R)$ is the irreducible representation of $SO(3)$ acting on the feature space, capturing both scalar and higher-order interactions.

**Patch Masking and Vector Quantization.** A fraction of patches are masked at a specified ratio $\rho$, and their embeddings are replaced by vector-quantized latents sampled from a learnable codebook $\mathcal{E}$ of size $N_B$. Following the discrete variational autoencoder (dVAE) relaxation, the quantized latent is computed as:

$$z_{p,i,m} = \frac{\sum_{j=1}^{N_B} \exp\left(\frac{g_j + \log q(e_j \mid h_{p,i,m})}{\tau}\right) e_j}{\sum_{j=1}^{N_B} \exp\left(\frac{g_j + \log q(e_j \mid h_{p,i,m})}{\tau}\right)}, \tag{32}$$

where $h_{p,i,m}$ is the hidden state, $e_j \in \mathcal{E}$ are codebook entries, $g_j \sim \text{Gumbel}(0, 1)$ is sampled noise, and $\tau$ is the softmax temperature. Visible and quantized embeddings are then passed to the decoder for reconstruction, preserving contextual information across masked regions.

**Reconstruction Targets.** For coordinate recovery, the decoder outputs:

$$\hat{X} = \text{Reshape}(\text{MLP}(H_p^{(L_2)})), \quad \hat{X} \in \mathbb{R}^{\delta \rho M \times K \times 3}, \tag{33}$$

where $H_p^{(L_2)}$ is the decoder output from the $L_2$-th layer, $M$ is the number of input molecular samples, and $\delta$ is a constant scaling factor. The multilayer perceptron (MLP) produces reconstructed Cartesian coordinates.

For surface geometry, each masked patch $X_{p,i,m}$ with center $x_{c,i,m}$ has covariance:

$$\Sigma = \tfrac{1}{K} \sum_{j=1}^{K} (x_{p,i,j,m} - x_{c,i,m})(x_{p,i,j,m} - x_{c,i,m})^\top \in \mathbb{R}^{3\times3}. \tag{34}$$

Pseudo-curvatures are derived from its eigenvalues $\epsilon_1, \epsilon_2, \epsilon_3$ as:

$$\psi_i = \left( \tfrac{\epsilon_1}{\epsilon_1+\epsilon_2+\epsilon_3}, \ \tfrac{\epsilon_2}{\epsilon_1+\epsilon_2+\epsilon_3}, \ \tfrac{\epsilon_3}{\epsilon_1+\epsilon_2+\epsilon_3} \right). \tag{35}$$

An MLP predicts $\hat{\psi}_i$, supervised by root mean square error (RMSE), ensuring accurate capture of local surface geometry.

**Training Objective.** The overall loss combines reconstruction, curvature, and regularization terms:

$$\mathcal{L} = \nu_1 \mathcal{L}_{\mathrm{rec}} + \nu_2 \mathcal{L}_{\mathrm{cur}} + \nu_3 \mathcal{L}_{\mathrm{KL}}, \tag{36}$$

where $\nu_1, \nu_2, \nu_3$ are scalar weights. The reconstruction loss uses the Chamfer distance:

$$\mathcal{L}_{\mathrm{rec}} = \frac{1}{\delta\rho M K} \sum_{i=1}^{\delta\rho M} \left( \sum_{a\in\hat{X}_i} \min_{b\in X_{p,i,m}} \|a-b\|_2^2 + \sum_{b\in X_{p,i,m}} \min_{a\in\hat{X}_i} \|a-b\|_2^2 \right). \tag{37}$$

The curvature loss is defined as:

$$\mathcal{L}_{\mathrm{cur}} = \frac{1}{\delta\rho M} \sum_{i=1}^{\delta\rho M} \|\psi_i - \hat{\psi}_i\|_2^2, \tag{38}$$

and the KL divergence term $\mathcal{L}_{\mathrm{KL}}$ regularizes the posterior $q(Z_{p,m} \mid H_{p,m})$ toward a uniform categorical prior.

This unsupervised pre-training framework leverages SE(3)-equivariant encoding, quantized masked reconstruction, and curvature supervision to learn robust docking-aware priors, providing an effective initialization for downstream fine-tuning.

### C.3 STAGE 3: SUPERVISED FINE-TUNING

In the third stage, the encoder pre-trained in Stage 2 is adapted to docking-specific tasks through supervised fine-tuning on protein–ligand complexes with ground-truth labels. This stage integrates cross-attention fusion to capture receptor–ligand interactions and employs three complementary supervision signals—pocket segmentation, interaction prediction, and binding affinity regression—to align representations with docking semantics. Below, we detail the computational framework and objectives, ensuring a seamless transition from encoding to multi-task learning.

**Equivariant Attention Fusion.** Given a receptor surface $R$ and a ligand $L$, their point-cloud representations are encoded independently by the Stage 2 encoders, yielding latent fields

$$Z_r \in \mathbb{R}^{N_r \times d}, \quad Z_l \in \mathbb{R}^{N_l \times d},$$

where $N_r$ and $N_l$ are the numbers of sampled points on the receptor and ligand surfaces, respectively, and $d$ is the embedding dimension. To model interfacial dependencies, we employ an SE(3)-equivariant attention mechanism, in which scalar attention weights are computed from $\ell = 0$ channels while higher-order features are rotated via Wigner-$D$ matrices to preserve equivariance:

$$\mathrm{Attn}(Z_r, Z_l) = \mathrm{softmax}\left( \frac{Q_r^{(\ell=0)}(K_l^{(\ell=0)})^\top}{\sqrt{d}} \right) V_l,$$

where $Q_r^{(\ell=0)} = Z_r^{(\ell=0)} W_Q$, $K_l^{(\ell=0)} = Z_l^{(\ell=0)} W_K$, and $V_l = Z_l W_V$ with $W_Q, W_K, W_V \in \mathbb{R}^{d\times d}$ being learnable projection matrices. The resulting cross-attended features are aggregated with the original receptor embeddings using a permutation-invariant operator $A(\cdot)$, implemented as the concatenation of mean- and max-pooling:

$$\tilde{z} = A(Z_r, \mathrm{Attn}(Z_r, Z_l)) \in \mathbb{R}^{2d}.$$

This fused representation captures both local and interfacial information, serving as the foundation for downstream supervision tasks.

**Multi-Task Supervision.** Three complementary supervised objectives align $\tilde{z}$ with docking semantics in a cascaded manner: the pocket prediction serves as a foundational gate, with its outputs weighting and conditioning the subsequent interaction and affinity predictions to emphasize the utility of the identified pocket. 1. *Pocket prediction*: Each receptor embedding $z_i \in Z_r$ (where $Z_r$ denotes the set of $N_r$ receptor embeddings) is classified as binding-site or non-binding-site, with ground-truth label $y_i \in \{0, 1\}$. The loss is defined as

$$\mathcal{L}_{\text{pocket}} = \frac{1}{N_r} \sum_{i=1}^{N_r} \text{BCE}(\hat{y}_i, y_i) + \lambda_p \, \mathcal{R}_{\text{geom}}, \tag{39}$$

where BCE denotes binary cross-entropy, $\hat{y}_i$ is the predicted probability, $N_r$ is the number of receptor embeddings, and $\lambda_p$ is a regularization weight. The geometric regularizer $\mathcal{R}_{\text{geom}}$ enforces agreement with distance-based pseudo-labels:

$$\mathcal{R}_{\text{geom}} = \frac{1}{N_r} \sum_{i=1}^{N_r} \|\hat{y}_i - y_i^{\text{geom}}\|^2,$$

where $y_i^{\text{geom}} = 1$ if the closest ligand point lies within a pre-defined cutoff radius, and $0$ otherwise. The predicted pocket probabilities $\hat{y}_i$ are used to gate subsequent losses.

2. *Interaction prediction*: Conditioned on the predicted pocket, the fused global vector $\tilde{z}$ (Eq. 10), processed through a multilayer perceptron (MLP) classifier, predicts whether the receptor–ligand pair forms a valid complex. The pocket-weighted loss is

$$\mathcal{L}_{\text{int}} = \frac{1}{N_r} \sum_{i=1}^{N_r} \hat{y}_i \cdot \text{BCE}(\hat{y}_{\text{int}}^{(i)}, y_{\text{int}}^{(i)}), \tag{40}$$

where $\hat{y}_{\text{int}}^{(i)} \in (0, 1)$ is the predicted interaction probability for the $i$-th receptor embedding (focusing on pocket regions), $y_{\text{int}}^{(i)} \in \{0, 1\}$ is the corresponding ground-truth label, $N_r$ is the number of receptor embeddings, and the weighting by $\hat{y}_i$ ensures emphasis on high-confidence pockets.

3. *Binding affinity regression*: For complexes with experimentally measured affinities, the model predicts binding free energy $\Delta G(r, l) \in \mathbb{R}$ (standardized to zero mean and unit variance), conditioned on both predicted pocket and interaction. The cascaded-weighted regression loss is

$$\mathcal{L}_{\Delta G} = \frac{1}{|\mathcal{V}|} \sum_{(r,l) \in \mathcal{V}} \left( \max(\hat{y}_i \cdot \hat{y}_{\text{int}}^{(i)}, \tau) \cdot \left( \hat{y}_{\Delta G}(r, l) - \Delta G(r, l) \right)^2 \right), \tag{41}$$

where $\mathcal{V}$ is the set of receptor–ligand pairs with ground-truth affinity values (with $|\mathcal{V}|$ denoting its cardinality), $\hat{y}_{\Delta G}(r, l)$ is the predicted binding free energy, $\hat{y}_i$ is the predicted pocket probability for the relevant receptor embedding $i$, $\hat{y}_{\text{int}}^{(i)}$ is the predicted interaction probability for that embedding, and $\tau$ is a minimum confidence threshold (e.g., 0.1). The weighting by $\hat{y}_i \cdot \hat{y}_{\text{int}}^{(i)}$ propagates the dependency from prior predictions, ensuring affinity regression focuses on viable pocket-based interactions.

**Overall Objective.** The joint fine-tuning objective combines the three tasks, each supervised by lightweight MLP heads:

$$\mathcal{L}_{\text{FT}} = \alpha \, \mathcal{L}_{\text{pocket}} + \beta \, \mathcal{L}_{\text{int}} + \mathcal{L}_{\Delta G}, \tag{42}$$

where $\alpha$ and $\beta$ are weighting coefficients. During optimization, the encoders (initialized from Stage 2), the cross-attention module, and task-specific MLP heads are jointly updated. This multi-task framework effectively couples receptor–ligand embeddings and specializes the pre-trained backbone for docking-aware representation learning, ensuring robust alignment with docking objectives.

## C.4 Stage 4: Inversion-based Ligand Generation

In the fourth stage, the docking-aware backbone is transformed into a generative engine via an inversion mechanism. Unlike sampling–ranking pipelines, inversion performs direct gradient-based

refinement in the latent continuous space, followed by projection and decoding into chemically valid discrete structures. This section provides the complete mathematical formulation, algorithms, and hyperparameters for ligand generation.

**Composite Objective.** Given a receptor $R$ and a candidate ligand $S$, their surfaces are independently encoded into latent fields $Z_r$ and $Z_l$ by the Stage 3 encoders. The fused interfacial embedding $\tilde{z}$ is obtained via cross-attention fusion (Eq. 10). The supervised heads trained in Stage 3 yield a differentiable composite objective function $\mathcal{F}$:

$$\mathcal{F}(S; R, \Theta) = \alpha\, \mathcal{L}_{\text{pocket}}(S; R) + \beta\, \mathcal{L}_{\text{int}}(S; R) + \mathcal{L}_{\Delta G}(S; R), \tag{43}$$

where $\mathcal{L}_{\text{pocket}}$ is the binary cross-entropy loss for binding pocket localization, $\mathcal{L}_{\text{int}}$ is the interaction plausibility loss (evaluating atom–residue interfacial compatibility), and $\mathcal{L}_{\Delta G}$ is the regression loss for binding free energy $\Delta G$. Here $\alpha = 1.0$ and $\beta = 0.5$ are balancing coefficients, and $\Theta$ are model parameters.

**Gradient-based Refinement.** Let $(\mathbf{x}, f)$ denote the differentiable ligand representation, where $\mathbf{x} \in \mathbb{R}^{N_l \times 3}$ are atom coordinates and $f \in \mathbb{R}^{N_l \times d_f}$ are atom features (including atomic type logits, partial charges, hybridization states, and hydrogen-bond polarity indicators). At each iteration $t$, the update step is

$$(\mathbf{x}, f)^{t+1} = \Pi_{\mathcal{C}_{\text{valid}}}\left((\mathbf{x}, f)^t - \eta_t \nabla_{(\mathbf{x}, f)} \mathcal{F}(S_t; R, \Theta)\right), \tag{44}$$

where $\eta_t = 10^{-3}$ is the step size, and $\Pi_{\mathcal{C}_{\text{valid}}}$ projects the updated state back onto the chemically and geometrically valid manifold $\mathcal{C}_{\text{valid}}$. Projection enforces valid valence, realistic bond lengths, and avoidance of steric clashes.

**Generative Mapping.** After projection, the refined continuous variables are decoded into chemically valid structures via a type-specific generative mapping $\mathcal{G}_{\text{type}}$. Unlike a deterministic argmax, $\mathcal{G}_{\text{type}}$ leverages gradient information to bias probabilistic sampling, followed by rule-based corrections to enforce validity.

For small molecules, atom typing is performed by softmax sampling:

$$a_i \sim \text{Categorical}(\sigma(f_i - \gamma \nabla_{f_i} \mathcal{F})), \quad a_i \in \mathcal{A}_{\text{atom}}, \tag{45}$$

where $\mathcal{A}_{\text{atom}} = \{\text{C(sp3), C(sp2), H, O(sp3), O(sp2), N(sp3), N(sp2), S(sp2)}\}$ is the atom-type set, $\sigma(\cdot)$ is the softmax function, and $\gamma = 0.1$ balances gradient bias.

Bond inference uses logits $g_{ij}$ and is jointly sampled over bond types $\{1, 2, 3, \text{aromatic}\}$:

$$b_{ij} \sim \text{Categorical}(\sigma(g_{ij} - \gamma \nabla_{g_{ij}} \mathcal{F})), \tag{46}$$

where bond types correspond to single, double, triple, and aromatic bonds. Feasibility is checked by distance thresholds (1.0–2.0 Å) and valence constraints $v_i \leq v_{\max}(a_i)$.

To incorporate structural motifs, we introduce a prior $\mathcal{P}_{\text{motif}}$ based on point cloud geometry, identifying centers via clustering of high-curvature, high-density points. Predefined templates guiding atom placement include: Benzene, Pyridine, Furan, Pyrrole, Thiophene, Imidazole, Carboxyl, Amide, etc. The motif probability is:

$$P_{\text{motif}}(x_i) \propto \exp\left(-\frac{\|\mathbf{x}_i - \mathbf{c}_{\text{motif}}\|^2}{\sigma_{\text{motif}}^2}\right) \cdot \Bbbk[\text{valid}(a_i, \mathbf{x}_i)], \tag{47}$$

where $\mathbf{c}_{\text{motif}}$ is the cluster center, $\sigma_{\text{motif}} = 0.5$ Å, and invalid structures are corrected by RDKit (a cheminformatics toolkit), with $\Delta G$ rewarding aromatic and functional motifs.

For protein ligands, residue identity $r_i$ is sampled similarly:

$$r_i \sim \text{Categorical}(\sigma(f_i - \gamma \nabla_{f_i} \mathcal{F})), \quad r_i \in \mathcal{A}_{20}, \tag{48}$$

where $\mathcal{A}_{20}$ is the canonical amino acid set. Backbone torsions $(\phi, \psi)$ are continuously updated and projected into $[-180°, 180°]$, while side-chain torsions $\chi_k$ are sampled from the Dunbrack rotamer library. Cartesian reconstruction is carried out using PyRosetta's internal geometry engine (a protein modeling suite), followed by energy minimization (Rosetta relax) to resolve steric clashes.

Thus, $\mathcal{G}_{\text{mol}}$ enforces atom- and bond-level chemical validity via stochastic decoding and sanitization, while $\mathcal{G}_{\text{prot}}$ performs residue-level sampling with biophysical torsional constraints. Both are tightly

coupled with gradient refinement, ensuring that sampled structures remain docking-aware while respecting chemical and geometric feasibility.

**Iterative Dynamics and Convergence.** Repeated application of gradient refinement and generative mapping yields a sequence of ligands $\{S_t\}$ converging to an optimized structure:

$$S^\star = \lim_{t \to T} \mathcal{G}_{\text{type}}\Big((\mathbf{x}, f)^t\Big), \tag{49}$$

with convergence typically observed within several hundreds steps. A cosine-annealed schedule is applied to $\eta_t \in [10^{-3}, 10^{-5}]$ to balance exploration and stability.

By tightly coupling gradient information with generative decoding, the framework ensures that optimization in latent space translates into chemically valid and docking-aware ligands. The distinction between $\mathcal{G}_{\text{mol}}$ (small molecules) and $\mathcal{G}_{\text{prot}}$ (proteins) allows the same inversion principle to adapt seamlessly to both types of ligands.

# D EXPERIMENT IN DETAILS

## D.1 COMPUTATION RESOURCE

All experiments were conducted on a single on-premise node unless otherwise specified. Table 3 summarizes the machine configuration and software stack to facilitate reproducibility.

Table 3: Compute environment used for all experiments.

| Component | Configuration |
|---|---|
| Server | Dell PowerEdge T640 |
| CPU | 2× Intel Xeon Gold 6240 @ 2.60 GHz (18 cores/socket; 36 cores, 72 threads total) |
| Memory | 251 GiB RAM |
| GPU | 4× NVIDIA A100 80GB PCIe (80 GiB each); driver 570.124.06; MIG disabled |
| GPU topology | GPU0/1 near NUMA node 0; GPU2/3 near NUMA node 1 |
| OS / Kernel | Ubuntu 22.04.5 LTS; Linux 5.15.0-126-generic |
| CUDA | Runtime 12.8 (from driver); Toolkit 12.4 (`nvcc`) |
| Python / Conda | Python 3.11.10 |
| DL stack | PyTorch 2.1.0 (cu118 build), torchvision 0.16.0, torchaudio 2.1.0 |

## D.2 DATASETS

**SKEMPI v2**( Liu et al. (2024)) is a mutation-centric benchmark of experimentally measured binding-affinity changes in protein–protein complexes, with receptor/ligand chains already labeled. Existing chain labels are retained (with spot corrections if needed). Entries without an affinity value are removed (57). As with other sources, affinities are converted to KD (M), the scope is restricted to PPIs, and duplicates are merged using the same Complex-ID scheme.

**Sabdab**( Liu et al. (2024)) is a large repository of antibody–antigen structures with explicit antigen, heavy-chain, and light-chain annotations. For consistency with a PPI setup, antigen chains are treated as receptor and antibody heavy/light chains as ligand. Records lacking affinity (14,148), non-PPI cases (46), entries missing antigen-chain labels (95), and chain-annotation errors (8) are removed. Remaining affinities are unified to KD (M), and duplicates across datasets are resolved with the Complex-ID definition above.

**PDBBind v2020**( Wang et al. (2004)) is a curated collection of biomolecular complexes; the protein–protein portion lists receptor/ligand names but not explicit chain IDs. Chain IDs are inferred with a semi-automatic procedure (parse chain descriptions, fuzzy-match to the annotated names) followed by expert proofreading and splitting of multi-complex PDBs, yielding chain assignments for 2,788 samples. Data cleaning removes non-PPI entries (6), records with ambiguous chain annotation (62), and entries whose reported affinities cannot be reliably converted to KD (62). All remaining affinities are standardized to KD (M), and cross-source duplicates are consolidated via a "Complex ID" composed of PDB code, sorted chains, mutations, and PubMed ID.

Table 4: The statistical information of the experimental datasets.

| Dataset | Samples | Complex Type | Affinity Info |
|---------|---------|--------------|---------------|
| SKEMPI v2 | 7,146 | Protein-Protein | KD, $\Delta G$ |
| Sabdab | 1,069 | Protein-Protein | KD, $\Delta G$ |
| PDBBind | 2,789 | Protein-Molecule | KD, $\Delta G$ |
| CrossDocked2020 | 18,450 | Protein-Molecule | pK |

**CrossDocked2020** ( Francoeur et al. (2020)) is a newly introduced dataset for structure-based machine learning, comprising 22.5 million poses of ligands docked into multiple similar binding pockets across the Protein Data Bank, designed to enhance the training and evaluation of grid-based convolutional neural network (CNN) models. This dataset includes cross-docked poses against non-cognate receptor structures and model-generated counterexamples, providing a standardized resource to recognize ligands in diverse target structures while significantly expanding the number of available poses for training.

### D.3 BASELINE MODELS

BASELINE MODELS FOR PROTEIN

**RAbD**( Adolf-Bryfogle et al. (2018)) introduces a knowledge-based Rosetta framework for computational antibody design that grafts canonical CDR loops from PyIgClassify, performs profile-guided sequence design with flexible-backbone sampling inside nested Monte-Carlo-plus-minimization cycles, and optimizes either total energy or interface $\Delta G$. However, its reliance on existing structural data may limit its effectiveness for designing antibodies against novel or poorly characterized antigens.

**DiffAb**( Luo et al. (2022)) proposes a diffusion-based, rotation/translation–equivariant generative model that co-designs antibody CDR sequences and 3D structures by iteratively denoising amino-acid types, $C\alpha$ coordinates, and SO(3) side-chain orientations, all explicitly conditioned on the target antigen structure. But its performance may be limited by the quality and diversity of the training data, potentially restricting its ability to generalize to a wide range of antigens.

**HSRN**( Jin et al. (2022)) introduces a hierarchical, rotation/translation–equivariant framework for antibody–antigen docking and design that combines a multi-scale encoder (atom- and residue-level) with an iterative, force-based refinement to fold and dock the paratope; during generation, an autoregressive decoder progressively docks partial paratopes and exploits the resulting geometric representation to choose the next residue. However, the model's heavy reliance on pre-defined epitope structures, assuming the input already provides the antigen's 3D structure and specific epitope location, poses a significant limitation, as epitope prediction remains a challenging task in practice, thereby restricting its end-to-end applicability.

**dyMEAN**( Kong et al. (2023)) [Kong et al., 2023] presents an end-to-end, E(3)-equivariant full-atom antibody design framework that, given an antigen epitope and an incomplete antibody sequence, initializes structure using conserved framework residues, attaches a "shadow paratope" to exchange invariant information and enable docking, and iteratively co-updates residue types and 3D coordinates with an adaptive multi-channel encoder that handles variable atom counts per residue; docking is finalized by aligning the native and shadow paratopes, and the method reports superior results on CDR-H3 generation, complex structure prediction, and affinity optimization. Although dyMEAN excels in end-to-end design and full-atom modeling, its limited scalability and reliance on high-quality training data remain limitations.

**AbX**( Zhu et al. (2024)) introduces a continuous-time, score-based diffusion framework that jointly models discrete CDR sequences (via a CTMC) and SE(3) coordinates, conditioned on the antigen/framework, and guided by evolutionary (ESM-2) priors plus geometric (FAPE, distogram, lDDT) and physical (violation, van der Waals) constraints to narrow the search space and improve binding/quality.

**ABDPO**( Zhou et al. (2024)) formulates antigen-specific sequence–structure co-design as direct energy-based preference optimization, fine-tuning a conditional diffusion model with residue-level energy preferences, decomposed attraction/repulsion terms, and gradient-surgery to resolve conflicts—thereby steering generations toward low total energy while maintaining binding affinity.

BASELINE MODELS FOR MOLECULE

**DrugFlow**( Schneuing et al. (2025)) introduces a generative framework for structure-based drug design that jointly models continuous ligand coordinates and discrete atom/bond types by combining Euclidean flow matching with discrete Markov bridges. The method extends to **FlexFlow**, which additionally samples protein side-chain torsion angles to capture binding-pocket flexibility. Key innovations include an end-to-end uncertainty estimator for out-of-distribution detection, a virtual node mechanism for adaptive molecule size selection, and a multi-domain preference alignment scheme that efficiently steers generation toward molecules with desirable drug-like properties. Experiments on CrossDocked demonstrate state-of-the-art distribution learning across chemical, geometric, and physical metrics. While DrugFlow excels in holistic distribution modeling and flexible sampling, challenges remain in scaling to larger protein systems and balancing preference alignment with sample validity.

**3D-SBDD**( Luo et al. (2021)) proposes a 3D generative framework for structure-based drug design that directly models ligand atoms within protein binding pockets using an autoregressive flow-based approach. The method conditions ligand generation on 3D protein environments, incrementally placing atoms while capturing geometric constraints, and employs equivariant neural networks to ensure rotational and translational invariance. Evaluation on CrossDocked shows significant improvements in binding pose accuracy, chemical validity, and docking performance compared to baseline methods. Despite its strong capability in geometry-aware ligand generation, challenges remain in handling larger, more flexible ligands and incorporating dynamic protein conformations.

**ALIDIFF**( Gu et al. (2024)) introduces a target-aware molecule diffusion framework that aligns generative sampling with exact energy optimization for structure-based drug design. The method integrates a diffusion backbone with a dual-stage alignment scheme: a coarse-grained alignment to enforce global docking plausibility and a fine-grained optimization that explicitly minimizes binding energies within the protein pocket. By coupling stochastic generative modeling with deterministic energy-based refinement, AliDiff achieves superior docking accuracy and binding affinity prediction on CrossDocked benchmarks. While it demonstrates strong target-conditioning and energy alignment, its reliance on accurate energy models and the computational overhead of fine-grained optimization pose scalability challenges.

**DiffSBDD**( Schneuing et al. (2024)) presents a diffusion-based generative framework for structure-based drug design that learns to directly sample 3D ligand structures conditioned on protein binding pockets. The model leverages SE(3)-equivariant neural networks to ensure rotational and translational invariance, and formulates ligand generation as a denoising process that progressively refines random atom clouds into chemically valid molecules docked in the target pocket. Extensive experiments on CrossDocked demonstrate improved performance over flow-based and autoregressive baselines in terms of pose accuracy, binding affinity, and chemical diversity. Despite its advantages in geometry-aware sampling, challenges remain in scalability to large ligands and efficient integration of protein flexibility.

**DockStream** ( Guo et al. (2021)) presents a flexible molecular docking wrapper that integrates with the de novo design platform REINVENT 2.0, providing access to various ligand embedders (e.g., Corina, LigPrep) and docking backends (e.g., AutoDock Vina, Glide) to enhance structure-based drug discovery by automating docking experiments, benchmarking configurations, and optimizing docking scores, while overcoming limitations of QSAR models through structural information; its scalability and performance vary by target, with ongoing challenges in accurately predicting binding free energies.

**liGAN** ( Ragoza et al. (2022)) introduces a deep learning system that generates 3D molecular structures conditioned on receptor binding sites using a conditional variational autoencoder trained on atomic density grids, employing atom fitting and bond inference to construct valid conformations, and demonstrates significant changes in generated molecules with mutated receptors; its reliance on high-quality structural data and computational complexity pose challenges for scalability.

## D.4 EVALUATION METRICS

### PROTEIN EVALUATION METRICS

**(1) IMP:** The Improvement Percentage (IMP) metric evaluates the relative enhancement in binding affinity achieved by designed protein sequences compared to their natural counterparts. Specifically, it quantifies the proportion of designed antibodies predicted to exhibit stronger binding than the corresponding natural antibodies, as assessed by the Rosetta interface energy function. Formally, IMP is defined as:

$$\text{IMP} = \frac{1}{N} \sum_{i=1}^{N} \mathbb{I}\left(E_i^{\text{design}} < E_i^{\text{natural}}\right) \times 100\%, \tag{50}$$

where $N$ is the total number of antibody–antigen pairs, $E_i^{\text{design}}$ denotes the Rosetta interface energy of the $i$-th designed antibody, $E_i^{\text{natural}}$ is the corresponding value for the natural antibody, and $\mathbb{I}(\cdot)$ is the indicator function returning 1 if the designed sequence has a lower (better) predicted binding energy than the natural sequence.

Higher IMP values indicate that a larger fraction of designed antibodies surpass their natural references in predicted binding affinity, highlighting the effectiveness of the design strategy in optimizing protein–protein interactions.

**(2) STA:** The Stability (STA) metric quantifies the conformational integrity and biochemical viability of designed proteins by integrating a weighted composite of steric hindrance assessment and structural coherence evaluation, thereby mitigating the risk of thermodynamically unstable or misfolded conformations. This metric is derived from two sub-components: the Steric Clash Score (SCS), which penalizes interatomic van der Waals overlaps indicative of steric repulsion, and the Secondary Structure Coherence (SSC), which evaluates the fidelity of predicted helical, sheet, and coil motifs against empirical folding propensities. Formally, STA is computed as:

$$\text{STA} = \frac{1}{N} \sum_{i=1}^{N} \left[\alpha \cdot \exp\left(-\frac{\text{SCS}(p_i)}{\sigma}\right) + \beta \cdot \left(1 - \frac{|\text{SSC}(p_i) - \mu|}{\tau}\right)^2\right], \tag{51}$$

where $N$ is the number of generated proteins, $p_i$ denotes the $i$-th protein, $\text{SCS}(p_i)$ and $\text{SSC}(p_i)$ represent the respective sub-scores derived from all-atom clash detection and dihedral angle consistency checks, $\alpha$ and $\beta$ are empirical weighting coefficients (with $\alpha + \beta = 1$), $\sigma$ and $\tau$ are scaling hyperparameters for normalization, and $\mu$ is the expected coherence baseline calibrated from native protein ensembles. In our implementation, the coefficient for secondary structure propensity ($\beta$) is set to 0.6, and the coefficient for collision detection ($\alpha$) is set to 0.4.

Elevated STA values signify enhanced adherence to biophysical constraints across the ensemble, underscoring the efficacy of the generative paradigm in yielding robust, functional protein architectures.

**(3) DIV:** The Diversity (DIV) metric quantifies the sequence and structural variability among generated antibodies, capturing the spread of designs in protein space. A diverse set of candidates increases the likelihood of discovering high-affinity binders with novel interaction profiles. Mathematically, DIV is defined as the average pairwise dissimilarity:

$$\text{DIV} = \frac{2}{N(N-1)} \sum_{i=1}^{N-1} \sum_{j=i+1}^{N} D(p_i, p_j), \tag{52}$$

where $N$ is the number of generated proteins, and $D(p_i, p_j)$ is a distance function that measures dissimilarity between proteins $p_i$ and $p_j$ in both sequence and structural space. Higher DIV values indicate a broader coverage of the protein design landscape, reducing redundancy in the generated set.

**(4) NOV:** The Novelty (NOV) metric measures how different the generated proteins are from the training dataset, considering both sequence and structural similarity. For a generated protein $p_i$ and a training protein $t_j$, the combined similarity score $S(p_i, t_j)$ is defined as:

$$S(p_i, t_j) = \alpha \cdot S_{\text{seq}}(p_i, t_j) + (1 - \alpha) \cdot S_{\text{str}}(p_i, t_j), \tag{53}$$

where $S_{\text{seq}}$ denotes sequence similarity, $S_{\text{str}}$ denotes structural similarity, and $\alpha \in [0, 1]$ controls their relative contribution. Given sequences of length $L$, we compute sequence identity as:

$$S_{\text{seq}}(p_i, t_j) = \frac{1}{L} \sum_{k=1}^{L} \mathbb{I}\big[p_i[k] = t_j[k]\big], \tag{54}$$

where $\mathbb{I}[\cdot]$ is the indicator function. For structural alignment, we use a TM-score–like normalized measure:

$$S_{\text{str}}(p_i, t_j) = \frac{1}{L} \sum_{k=1}^{L} \frac{1}{1 + \left(\frac{d_k}{d_0}\right)^2}, \tag{55}$$

where $d_k$ is the distance between aligned $C_\alpha$ atoms, and $d_0$ is a length-dependent normalization constant. For each generated protein $p_i$, we take the maximum similarity across all training proteins and define novelty as:

$$\text{NOV} = 1 - \frac{1}{N} \sum_{i=1}^{N} \max_{t_j \in \mathcal{T}} S(p_i, t_j), \tag{56}$$

where $N$ is the number of generated proteins and $\mathcal{T}$ is the training set. Higher NOV values indicate stronger novelty in both sequence and structure.

**(5) AAR:** The Amino Acid Recovery (AAR) metric measures the sequence-level accuracy of generated protein sequences by comparing them against their corresponding native (reference) sequences. This metric reflects how well the generative model is able to reproduce the original amino acid composition, thereby serving as an indicator of sequence fidelity and preservation of native biochemical properties. The mathematical formulation is given by:

$$\text{AAR} = \frac{1}{N \cdot L} \sum_{i=1}^{N} \sum_{j=1}^{L} \mathbb{I}\big(a_{ij}^{\text{gen}} = a_{ij}^{\text{ref}}\big) \times 100\%, \tag{57}$$

where $N$ is the number of protein sequences, $L$ is the sequence length, $a_{ij}^{\text{gen}}$ denotes the amino acid at position $j$ in the $i$-th generated sequence, $a_{ij}^{\text{ref}}$ is the corresponding amino acid in the native sequence, and $\mathbb{I}(\cdot)$ is the indicator function.

Higher AAR values indicate closer agreement with native sequences, suggesting that the generated proteins better retain structural and functional characteristics inherent to the original proteins.

MOLECULE EVALUATION METRICS

**(1) Vina Score:** The Vina Score metric provides an estimate of the binding affinity between a generated small molecule (ligand) and a target protein. It reflects the stability of the protein–ligand complex predicted during molecular docking. Following the formulation implemented in **AutoDock Vina** , the score approximates the binding free energy based on an empirical scoring function that accounts for key interaction terms, including steric complementarity, hydrogen bonding, hydrophobic interactions, and torsional entropy penalties. The mathematical formulation is given by:

$$E_{\text{Vina}} = E_{\text{gauss}} + E_{\text{repulsion}} + E_{\text{hydrophobic}} + E_{\text{hydrogen}} + E_{\text{torsional}}, \tag{58}$$

where $E_{\text{gauss}}$ models attractive van der Waals interactions, $E_{\text{repulsion}}$ penalizes steric clashes, $E_{\text{hydrophobic}}$ captures hydrophobic contacts, $E_{\text{hydrogen}}$ measures hydrogen bond formation, and $E_{\text{torsional}}$ accounts for the conformational entropy cost of ligand flexibility.

The resulting Vina Score is reported in kcal/mol, with more negative values indicating stronger predicted binding affinities. Typically, scores range from around $-4$ kcal/mol (weak binding) to below $-10$ kcal/mol (highly favorable binding). Although not a direct physical free energy, the score serves as a comparative metric to rank ligands by their likelihood of stable binding.

**(2) High-affinity:** The High-affinity metric quantifies the proportion of generated molecules that exhibit stronger predicted binding to a given protein target than their corresponding reference ligands. Binding strength is assessed using Vina Scores, where lower (more negative) values indicate higher predicted affinity. The mathematical formulation is given by:

$$\text{High-affinity} = \frac{1}{N} \sum_{i=1}^{N} \mathbb{I}\big(E_i^{\text{gen}} < E_i^{\text{ref}}\big), \tag{59}$$

where $N$ denotes the total number of generated molecules, $E_i^{\text{gen}}$ is the Vina Score of the $i$-th generated molecule, $E_i^{\text{ref}}$ is the Vina Score of its corresponding reference ligand, and $\mathbb{I}(\cdot)$ is the indicator function returning 1 if the condition is satisfied and 0 otherwise.

The metric outputs the fraction of molecules surpassing the reference ligands in predicted affinity, providing a normalized measure of how frequently the generation process yields candidates with potentially improved binding properties.

**(3) STA:** The Stability (STA) metric for small-molecule generation evaluates the chemical plausibility and conformational robustness of designed ligands by integrating synthetic accessibility and conformational strain energy into a unified score. This ensures that generated compounds are not only geometrically valid but also chemically feasible under standard synthesis and physiological conditions. Specifically, the metric combines two sub-components: the Synthetic Accessibility Index (SAI), which penalizes ligands with rare or chemically intractable substructures, and the Conformational Strain Energy (CSE), which quantifies the internal energetic penalty required to maintain a given 3D geometry relative to its energy-minimized conformation. Formally, STA is defined as:

$$\text{STA} = \frac{1}{M} \sum_{j=1}^{M} \left[ \gamma \cdot \exp\left( -\frac{\text{CSE}(m_j)}{\lambda} \right) + \delta \cdot \left( 1 - \frac{\text{SAI}(m_j) - \mu}{\kappa} \right)^2 \right], \tag{60}$$

where $M$ is the number of generated molecules, $m_j$ denotes the $j$-th molecule, $\text{CSE}(m_j)$ is the strain energy computed via molecular mechanics force fields, and $\text{SAI}(m_j)$ is a normalized synthetic accessibility score. $\gamma$ and $\delta$ are weighting coefficients with $\gamma + \delta = 1$.

In our implementation, the scaling parameter is fixed as $\lambda = 10$, the normalization parameter as $\kappa = 1.5$, and the baseline accessibility as $\mu = 3.0$, the coefficient for SAI ($\gamma$) is set to 0.5, and the coefficient for collision detection ($\delta$) is set to 0.5. These hyperparameters are selected to provide a stable balance between strain minimization and synthetic feasibility.

**(4) DIV:** The Top-K Diversity (DIV) metric quantifies the structural diversity of the top-K generated molecules, reflecting the spread of chemical structures within a given set. Following the methodology of (Bengio et al. (2021)), DIV is defined as the average pairwise Tanimoto distance between the Morgan fingerprints of the generated molecules. The mathematical formulation is given by:

$$\text{DIV} = \frac{2}{K(K-1)} \sum_{i=1}^{K-1} \sum_{j=i+1}^{K} \text{Tanimoto}(FP_i, FP_j), \tag{61}$$

where $K$ is the number of top-ranked molecules (e.g., top-10), $FP_i$ and $FP_j$ are the Morgan fingerprint vectors for molecules $i$ and $j$, and the Tanimoto similarity is computed as:

$$\text{Tanimoto}(FP_i, FP_j) = \frac{|FP_i \cap FP_j|}{|FP_i \cup FP_j|}, \tag{62}$$

with $|FP_i \cap FP_j|$ and $|FP_i \cup FP_j|$ representing the number of common and total unique bits, respectively. The Tanimoto distance is then $1 - \text{Tanimoto}(FP_i, FP_j)$, ensuring a range of 0 (identical structures) to 1 (completely dissimilar structures). The factor $\frac{2}{K(K-1)}$ normalizes the average over all unique pairs.

**(5) NOV:** Following InversionGNN ( Niu et al. (2025)), the Novelty (NOV) metric quantifies the proportion of generated molecules that are not present in the training set, serving as an indicator of the model's ability to explore beyond the learned chemical space. This metric is particularly relevant in de novo molecular design, where generating novel structures is crucial for discovering new drug candidates. The mathematical formulation is given by:

$$\text{Nov} = \frac{N_{\text{new}}}{N_{\text{total}}}, \tag{63}$$

where $N_{\text{new}}$ is the number of generated molecules that do not appear in the training set, and $N_{\text{total}}$ is the total number of generated molecules evaluated. The value of Nov ranges from 0 to 1, with higher values indicating a greater proportion of novel molecules.

## D.5 MORE EXPERIMENT FIGURES & TABLES

### D.5.1 ABLATION STUDY

For the ablation experiments of Stage 1, Stage 2, and Stage 3, each module plays an indispensable role in the generation of the final ligand. In addition, the ablation experiment for stage 4 showed that compared to the exhaustive method, MagicDock's gradient based selective generation strategy has an order of magnitude efficiency advantage while maintaining a basically consistent effect, proving the pertinence of Stage 4.

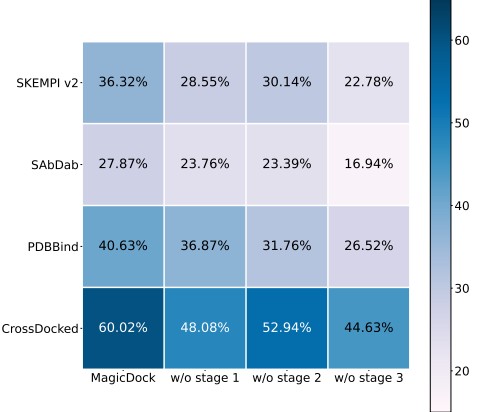

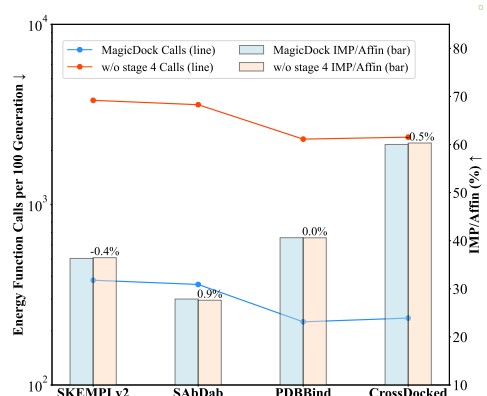

(a) Performance comparison between models with and without stage 1,2 and 3.

(b) Performance and cost comparison between models with and without stage 4.

Figure 6: Ablation study on the impact of different stages in MagicDock.

### D.5.2 GEOMETRIC AND FEATURE NOISE STUDY

We evaluate robustness under structural perturbations on $N = 100$ receptor–ligand pairs by applying Gaussian coordinate noise and feature dropout to receptor surfaces. Ligand generation are repeated on noisy vs. clean inputs, and metrics are reported in Fig. 7.

For protein ligands, robustness is measured by IMP, AAR, and RMSD. MagicDock achieves higher IMP and lower RMSD compared with DiffAb and Abx, demonstrating its stability under noise. For small molecule ligands, robustness is assessed using Vina score, High-affinity, and QED. Magic-Dock consistently yields more negative Vina scores and higher high-affinity rates, indicating resilience to input perturbations. Together, these results confirm that MagicDock maintains reliable performance under realistic noise, highlighting robustness to biological uncertainty.

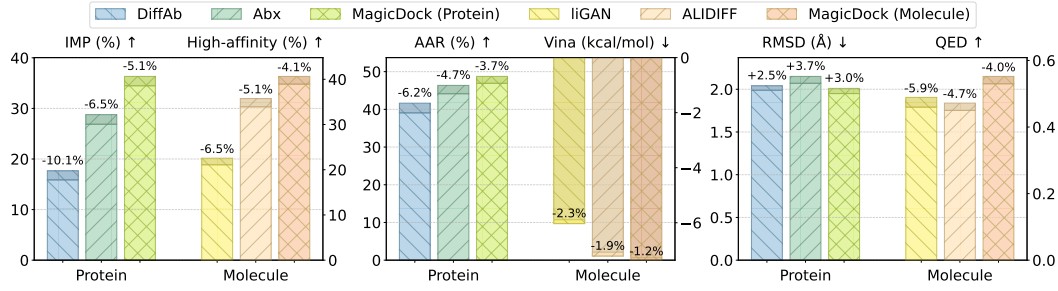

Figure 7: Robustness under geometric and feature noise, evaluated on IMP/High-affinity (%), AAR/Vina (kcal/mol), and RMSD/QED respectively on protein and molecular baselines.

### D.5.3 LOCAL PERTURBATION CONSISTENCY STUDY

Finally, we evaluate whether gradient attributions faithfully predict the energetic consequences of local structural edits. We introduce small geometric and chemical perturbations around high-attribution sites and compare gradient-predicted energy changes with the actual $\Delta\mathcal{F}$ measured after re-evaluation. This experiment directly tests whether attribution scores can serve as actionable signals for structural refinement, beyond passive localization. Performance is assessed by the sign-consistency rate (SCR), the coefficient of determination ($R^2$), rank correlation (Spearman $\rho$), and the mean observed energy change $\Delta\mathcal{F}$, where negative values indicate improved binding. As shown in Fig. 8, our IG-guided strategy substantially outperforms saliency and Grad×Input baselines, and approaches the oracle behavior of physics-based Rosetta evaluations.

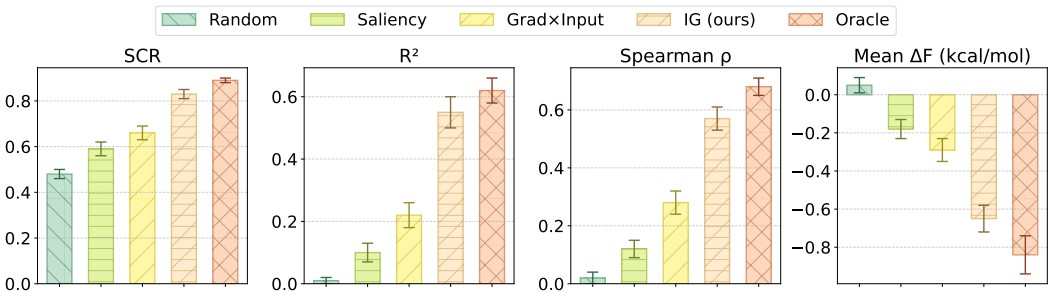

Figure 8: Evaluation of local perturbation consistency, showing SCR, R², Spearman correlation, and Mean $\Delta$F (kcal/mol) for attribution methods.

### D.5.4 CONFORMATIONAL AND MUTATIONAL VARIABLITY STUDY

We further evaluate remarkable robustness to biological variability by subjecting $N = 100$ receptors to perturbations, including conformer ensembles, single-point mutations, and combined variants. These perturbed structures are systematically processed through the inversion pipeline. Performance degradation is assessed via IMP for protein-based methods and High-Affinity for small-molecule methods, with MagicDock reporting both metrics. As shown in Fig. 9, statistical tests compare conservative and non-conservative mutations, illustrating MagicDock's exceptional robustness. MagicDock demonstrates minimal degradation and consistently superior performance over baselines.

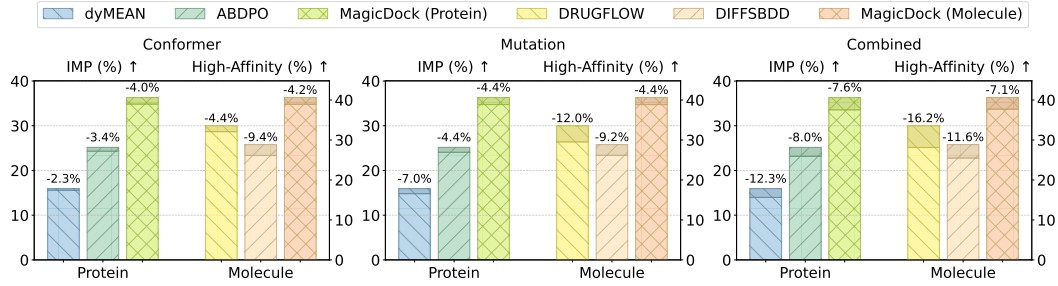

Figure 9: Robustness to conformational and mutational variability.

### D.5.5 RUNTIME AND RESOURCE USAGE STUDY

We benchmark MagicDock against baselines using 100 receptors, generating one ligand per receptor under identical hardware. Wall-clock time and memory usage are assessed per computational stage. Fig. 10a demonstrates MagicDock's reduced runtime while Fig. 10b highlights lower peak memory consumption, underscoring its resource efficiency.

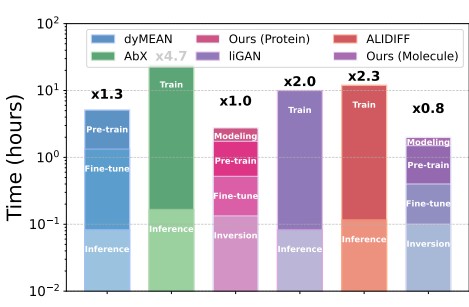
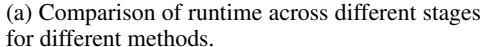
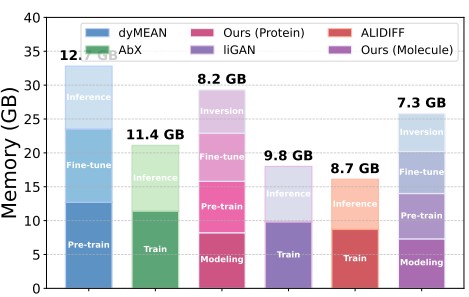

(a) Comparison of runtime across different stages for different methods.

(b) Comparison of Peak memory usage (GB) of different methods.

Figure 10: Comparison of runtime and resource usage of different methods.

### D.5.6 SCALABILITY STUDY

Scalability is evaluated across nine settings varying receptor sizes (500, 2000, 5000 points) and ligand complexities (30, 80, 150 atoms/residues) using 100 receptors from CrossDocked2020 and SAbDab. Metrics include per-iteration latency, iterations-to-converge, memory footprint, and scaling exponent $\gamma$ from $T \propto N^\gamma$.

Table 5: Scalability results across nine complexity settings. Values are averages over three runs, with Magic-Dock/dyMEAN/ALIDIFF separated by '/'.

| Receptor | Ligand | Latency (s/iter) | Iterations | Memory (GB) |
|---|---|---|---|---|
| Small | Low | 0.42/1.05/0.58 | 45/215/380 | 7.6/11.2/8.1 |
| | Medium | 0.65/1.35/0.85 | 108/285/455 | 8.4/12.8/9.2 |
| | High | 0.78/1.55/1.12 | 182/358/590 | 9.3/13.9/10.3 |
| Medium | Low | 1.15/2.25/1.18 | 48/335/440 | 10.3/16.8/11.6 |
| | Medium | 1.36/2.85/1.65 | 112/372/595 | 11.7/18.5/13.2 |
| | High | 1.72/3.45/2.05 | 192/445/545 | 12.8/20.8/14.7 |
| Large | Low | 2.28/5.60/2.75 | 58/405/480 | 15.9/26.5/17.8 |
| | Medium | 2.75/6.95/3.85 | 125/475/640 | 18.2/31.5/20.2 |
| | High | 3.35/8.90/4.75 | 245/478/780 | 20.5/36.2/23.8 |

Table 6: Fitted runtime exponents $\gamma$ for each method. This table evaluates the computational efficiency of different ligand design methods by presenting their fitted runtime exponents $\gamma$, which quantify how runtime scales with input size in a polynomial manner

| Method | $\gamma$ |
|---|---|
| MagicDock | 1.4 |
| dyMEAN | 1.8 |
| ALIDIFF | 1.7 |

## E THEORETICAL ANALYSIS IN DETAILS

### E.1 PROOF OF SE(3)-EQUIVARIANCE FOR MAGICDOCK

We analyze the SE(3)-equivariance of MagicDock under idealized assumptions. While the practical implementation may include minor numerical deviations (e.g., floating-point tie-breaking, discretization), Stages 1–3 are implemented with strictly equivariant modules, and Stage 4 is approximately equivariant due to non-convex chemical validity constraints. The following proof shows that the framework is SE(3)-equivariant in the limit of exact equivariant modules and convex surrogates.

**Theorem 1** (SE(3)-Equivariance of MagicDock). *Given the receptor point cloud $P_{rec}$ and the initial ligand point cloud $P_{lig}^0$, let the optimized ligand be $P_{lig}^* = \text{MagicDock}(P_{rec}, P_{lig}^0)$. Under the assumptions that (i) the chemical validity set $\mathcal{C}_{valid}$ admits a convex surrogate and exact Euclidean projection, (ii) all learned modules are SE(3)-equivariant, and (iii) features are restricted to invariant or properly transformed irreducible representations, MagicDock is SE(3)-equivariant. Namely, for any $g \in \text{SE}(3)$,*

$$g \cdot P_{lig}^* = \text{MagicDock}(g \cdot P_{rec}, g \cdot P_{lig}^0),$$

*where $g \cdot P := RP + t$ for $R \in \mathrm{SO}(3)$, $t \in \mathbb{R}^3$ acts on coordinates, while scalar features remain invariant.*

**Lemma 1.1** (Stage 1: Surface Point Cloud Modeling). *The surface point cloud mapping $\mathcal{S}$ is SE(3)-equivariant: $\mathcal{S}(g \cdot X) = g \cdot \mathcal{S}(X)$.*

*Proof.* Since the smoothed distance function (SDF) and associated weights depend only on Euclidean distances, which are invariant under rotations and translations, we have $\mathrm{SDF}(Rx + t) = \mathrm{SDF}(x)$. Mesh generation steps (gradient descent projection, FPS, KNN) depend only on pairwise distances and thus commute with $g$. **In practice:** deterministic FPS/KNN and numerically stable SDF are implemented, so equivariance holds up to floating-point error. □

**Lemma 1.2** (Stage 2: Pre-training with Equivariant Encoder). *The SE(3)-equivariant backbone $f_\theta$ built from tensor field convolutions is SE(3)-equivariant.*

*Proof.* The convolution kernels are of the form

$$z_i = \sum_{j \in \mathcal{N}(i)} \sum_{l=0}^{L} R_l(\|\mathbf{x}_{ij}\|) Y_l\left(\frac{\mathbf{x}_{ij}}{\|\mathbf{x}_{ij}\|}\right) \cdot W_l f_j,$$

which are equivariant because (i) radial functions depend only on invariant distances, and (ii) spherical harmonics transform according to irreducible representations of $\mathrm{SO}(3)$. **In practice:** the encoder is implemented with the `e3nn` library, which guarantees strict SE(3)-equivariance. □

**Lemma 1.3** (Stage 3: Supervised Fine-tuning with Equivariant Attention). *When scalar attention weights are computed from $\ell = 0$ channels and higher-order features are rotated via Wigner-D matrices, the cross-attention layers preserve SE(3)-equivariance.*

*Proof.* Attention scores $A$ computed from $\ell = 0$ channels are invariant. Values $V^{(\ell)}$ transform by $\rho^{(\ell)}(R)$ and aggregation

$$\tilde{Z}_r^{(\ell)} = \sum_j A_{ij}\, \rho^{(\ell)}(R)\, V_{l,j}^{(\ell)}$$

yields $\tilde{Z}_r^{(\ell)}(g \cdot P) = \rho^{(\ell)}(R)\, \tilde{Z}_r^{(\ell)}(P)$. **In practice:** we explicitly use irreducible representations and Wigner-$D$ matrices in attention, so equivariance is preserved exactly. □

**Lemma 1.4** (Stage 4: Inversion-based Generation). *The iterative update*

$$P_{lig}^{t+1} = \Pi_{\mathcal{C}_{valid}}\left(P_{lig}^t - \eta_t \nabla_{P_{lig}} \mathcal{F}(P_{lig}^t; P_{rec}, \Theta)\right),$$

*is SE(3)-equivariant under convex surrogate constraints.*

*Proof.* If $\mathcal{F}$ is built from invariant quantities (distances, angles), then $\mathcal{F}(g \cdot P_{\mathrm{lig}}; g \cdot P_{\mathrm{rec}}) = \mathcal{F}(P_{\mathrm{lig}}; P_{\mathrm{rec}})$. By the chain rule, for $y = Rx + t$ we have

$$\nabla_y \mathcal{F}(y) = R \nabla_x \mathcal{F}(x),$$

since $\delta\mathcal{F} = \nabla_x \mathcal{F} \cdot \delta x = \nabla_y \mathcal{F} \cdot \delta y$ with $\delta y = R\delta x$. Thus gradient steps commute with $g$:

$$g \cdot \left(P - \eta\nabla_P \mathcal{F}\right) = (RP + t) - \eta(R\nabla_P \mathcal{F}) = g \cdot P - \eta\nabla_{g \cdot P} \mathcal{F}.$$

If $\mathcal{C}_{\mathrm{valid}}$ is convex and invariant, projection also preserves equivariance. **In practice:** the true chemical validity set is highly non-convex (bond formation, aromaticity, topology constraints), so we enforce validity via heuristic repair. This makes Stage 4 only approximately equivariant in realistic settings. □

*Proof of Theorem.* Each stage (surface modeling, equivariant encoder, equivariant attention, inversion) preserves SE(3)-equivariance under the stated assumptions. By closure of equivariant maps under composition, MagicDock is SE(3)-equivariant in the idealized setting. □

**Remark on Practical Implementations.** In our implementation: Stage 1–Stage 3 are strictly SE(3)-equivariant (up to numerical precision) as guaranteed by design choices (deterministic SDF/FPS/KNN, e3nn-based encoder, Wigner-$D$ based attention). Stage 4 involves chemical validity projection on a non-convex manifold and is therefore only approximately equivariant. For reproducibility, we use deterministic tie-breaking in FPS/KNN, fixed random seeds, and numerically stable softmax. Thus MagicDock should be regarded as *exactly SE(3)-equivariant in Stages 1–3* and *approximately equivariant in Stage 4*.

### E.2 PROOF OF CONVERGENCE FOR MAGICDOCK

We analyze the convergence of the gradient-driven inversion stage in MagicDock under idealized assumptions. The analysis follows projected gradient descent (PGD) theory in non-convex optimization, while clarifying the role of convex approximations of the chemical validity manifold.

**Theorem 2** (PGD Convergence under Convex Approximation). *Let $\mathcal{F}(P_{lig}; P_{rec}, \Theta)$ be differentiable, $L$-smooth, and bounded below by $\mathcal{F}^* > -\infty$. Assume the validity constraint set $\mathcal{C}_{valid}$ is nonempty, closed, bounded, and convex (serving as an idealized surrogate for chemical validity). For a fixed step size $0 < \eta \leq 1/L$, consider the iteration*

$$P_{lig}^{t+1} = \Pi_{\mathcal{C}_{valid}}\left(P_{lig}^t - \eta \nabla \mathcal{F}(P_{lig}^t)\right),$$

*where $\Pi_{\mathcal{C}_{valid}}$ denotes Euclidean projection. Then:*

1. *The objective decreases monotonically:*

$$\mathcal{F}(P_{lig}^{t+1}) \leq \mathcal{F}(P_{lig}^t) - \frac{\eta}{2}\|g_\eta(P_{lig}^t)\|^2,$$

   *where $g_\eta(x) = \frac{1}{\eta}\left(x - \Pi_{\mathcal{C}_{valid}}(x - \eta \nabla \mathcal{F}(x))\right)$ is the gradient mapping.*

2. *The sum of squared gradient mappings is bounded:*

$$\sum_{t=0}^{T-1} \|g_\eta(P_{lig}^t)\|^2 \leq \frac{2}{\eta}\left(\mathcal{F}(P_{lig}^0) - \mathcal{F}^*\right).$$

3. *Consequently, $\liminf_{t\to\infty} \|g_\eta(P_{lig}^t)\| = 0$, and every accumulation point $P^*$ satisfies $g_\eta(P^*) = 0$, i.e., it is a first-order stationary point of $\mathcal{F}$ over $\mathcal{C}_{valid}$.*

**Lemma 2.1** (Descent Lemma for PGD). *For $L$-smooth $\mathcal{F}$ and $\eta \leq 1/L$, the PGD update $x^+ = \Pi_{\mathcal{C}}(x - \eta \nabla \mathcal{F}(x))$ satisfies*

$$\mathcal{F}(x^+) \leq \mathcal{F}(x) - \frac{\eta}{2}\|g_\eta(x)\|^2,$$

*where $g_\eta(x) = \frac{1}{\eta}(x - x^+)$.*

*Proof.* Let $y = x - \eta \nabla \mathcal{F}(x)$ and $x^+ = \Pi_{\mathcal{C}}(y)$. By $L$-smoothness:

$$\mathcal{F}(x^+) \leq \mathcal{F}(x) + \langle \nabla \mathcal{F}(x), x^+ - x \rangle + \frac{L}{2}\|x^+ - x\|^2.$$

By projection optimality, $(y - x^+)^\top (z - x^+) \leq 0$ for all $z \in \mathcal{C}$. Choosing $z = x$ and expanding $y = x - \eta \nabla \mathcal{F}(x)$ gives

$$\langle \nabla \mathcal{F}(x), x^+ - x \rangle \leq -\frac{1}{\eta}\|x^+ - x\|^2.$$

Substitute into the smoothness bound:

$$\mathcal{F}(x^+) \leq \mathcal{F}(x) - \frac{1}{\eta}\|x^+ - x\|^2 + \frac{L}{2}\|x^+ - x\|^2.$$

Since $g_\eta(x) = \frac{1}{\eta}(x - x^+)$ and $\eta \leq 1/L$, we obtain

$$\mathcal{F}(x^+) \leq \mathcal{F}(x) - \frac{\eta}{2}\|g_\eta(x)\|^2.$$

$\square$

**Lemma 2.2** (Bounded Sum of Gradient Mappings). *Summing Lemma 2.4 over $T$ iterations yields*

$$\sum_{t=0}^{T-1} \|g_\eta(P_{lig}^t)\|^2 \leq \tfrac{2}{\eta}(\mathcal{F}(P_{lig}^0) - \mathcal{F}^*).$$

*Proof.* Telescoping $\mathcal{F}(P_{\text{lig}}^t) - \mathcal{F}(P_{\text{lig}}^{t+1}) \geq \tfrac{\eta}{2}\|g_\eta(P_{\text{lig}}^t)\|^2$ gives

$$\mathcal{F}(P_{\text{lig}}^0) - \mathcal{F}(P_{\text{lig}}^T) \geq \tfrac{\eta}{2}\sum_{t=0}^{T-1} \|g_\eta(P_{\text{lig}}^t)\|^2.$$

Since $\mathcal{F}(P_{\text{lig}}^T) \geq \mathcal{F}^*$, the bound follows. $\square$

**Lemma 2.3** (Stationary Point Convergence). *The sequence $\{P_{lig}^t\}$ is bounded and has accumulation points. Every accumulation point $P^*$ satisfies $g_\eta(P^*) = 0$, i.e., $P^*$ is a stationary point satisfying first-order optimality conditions.*

*Proof.* Because $\mathcal{C}_{\text{valid}}$ is closed and bounded, the iterates remain in a compact set, ensuring accumulation points exist. From Lemma 2.2, $\liminf_{t\to\infty} \|g_\eta(P_{\text{lig}}^t)\| = 0$. By continuity of $g_\eta$, any limit point satisfies $g_\eta(P^*) = 0$. $\square$

*Proof of Theorem 2.* Lemma 2.4 establishes monotonic descent. Lemma 2.2 bounds the cumulative gradient mapping norms. Lemma 2.3 implies every accumulation point is stationary. Thus the PGD sequence converges to stationary points under the convex surrogate assumption. $\square$

**Remark on Non-Convex Validity Manifold.** In practice, the chemical validity set $\mathcal{C}_{\text{valid}}$ is highly non-convex due to valency, aromaticity, and stereochemistry constraints. The above proof holds only under the convex surrogate assumption, which serves as an *idealization*. When using generative validity repair (e.g., $\mathcal{G}_{\text{type}}$) instead of exact projection, the iteration behaves as an *inexact projection method*. Standard results on inexact PGD imply convergence to an $\varepsilon$-stationary point, with $\varepsilon$ depending on the repair error. Thus the theoretical guarantee should be interpreted as *asymptotic convergence under convex surrogates*, while practical implementations achieve only *approximate stationarity*.

### E.3 PROOF OF THEORETICAL SUPERIORITY FOR INVERSION FRAMEWORK

We compare the reachable objective values produced by (i) the common two-stage *generate-then-optimize* pipeline (G+O) and (ii) the gradient-driven *inversion* framework used in MagicDock. Let $(\mathcal{X}, \|\cdot\|)$ be a Euclidean space of parametrized structures (e.g., point clouds, coordinates, or features), and let

$$\mathcal{C}_{\text{valid}} \subseteq \mathcal{X}$$

denote the feasible set of chemically valid structures. We assume $\mathcal{C}_{\text{valid}}$ is closed. For analysis, we assume it is convex or that a convex surrogate is available; for nonconvex cases, we use a proximity operator (see Remark E.3.2). Let the design objective be

$$\mathcal{F} : \mathcal{C}_{\text{valid}} \to \mathbb{R},$$

which is differentiable, bounded below by $\mathcal{F}^\star > -\infty$, and $L$-smooth:

$$\|\nabla\mathcal{F}(x) - \nabla\mathcal{F}(y)\| \leq L\|x - y\|, \quad \forall x, y \in \mathcal{C}_{\text{valid}}.$$

We model the two paradigms as follows.

#### E.3.1 GENERATE-THEN-OPTIMIZE (G+O)

A generator $\mathcal{G} : \mathcal{Z} \to \mathcal{C}_{\text{valid}}$ (with latent prior $z \sim \mu_Z$) outputs $x_0 = \mathcal{G}(z)$. A local optimizer $\mathcal{O}$, defined as a short-step projected gradient descent (PGD) with step size $\eta \leq 1/L$, maps $x_0$ to a refined point $x_\infty = \mathcal{O}(x_0)$. The G+O reachable set is

$$\mathcal{R}_{\text{G+O}} = \{\mathcal{O}(\mathcal{G}(z)) : z \in \text{supp}(\mu_Z)\}.$$

### E.3.2 INVERSION (GRADIENT-DRIVEN)

Starting from an initialization $x_0 \in \mathcal{C}_{\text{valid}}$, inversion performs projected gradient-like updates:

$$x_{t+1} = \Pi_{\mathcal{C}_{\text{valid}}}\big(x_t - \eta_t \nabla \mathcal{F}(x_t)\big), \quad 0 < \eta_t \leq 1/L, \tag{64}$$

where $\Pi_{\mathcal{C}_{\text{valid}}}$ denotes the Euclidean projection onto $\mathcal{C}_{\text{valid}}$. Let $\mathcal{R}_{\text{Inv}}$ be the set of accumulation points of sequences generated by equation 64 initialized from all possible $x_0 \in \mathcal{C}_{\text{valid}}$ (or from $\mathcal{G}(z)$). We show: (i) inversion satisfies standard descent and convergence guarantees, (ii) $\mathcal{R}_{\text{G+O}} \subseteq \mathcal{R}_{\text{Inv}}$, and (iii) when the generator $\mathcal{G}$ is *support-misspecified*, inversion achieves strictly better infimal objective values.

**Lemma 2.4** (Descent for Projected Gradient Updates). *Under the above assumptions, the update equation 64 with constant step size $\eta \in (0, 1/L]$ satisfies for every $t$:*

$$\mathcal{F}(x_{t+1}) \leq \mathcal{F}(x_t) - \frac{\eta}{2}\|g_\eta(x_t)\|^2,$$

*where $g_\eta(x_t) = \frac{1}{\eta}\big(x_t - \Pi_{\mathcal{C}_{\text{valid}}}(x_t - \eta \nabla \mathcal{F}(x_t))\big)$ is the gradient mapping. Consequently,*

$$\sum_{t=0}^{T-1} \|g_\eta(x_t)\|^2 \leq \frac{2}{\eta}\big(\mathcal{F}(x_0) - \mathcal{F}^\star\big),$$

*and hence $\min_{0 \leq t < T} \|g_\eta(x_t)\|^2 \to 0$ as $T \to \infty$.*

*Proof.* Let $\tilde{x}_{t+1} := x_t - \eta \nabla \mathcal{F}(x_t)$ be the unconstrained gradient step, and $x_{t+1} = \Pi_{\mathcal{C}_{\text{valid}}}(\tilde{x}_{t+1})$. By $L$-smoothness,

$$\mathcal{F}(y) \leq \mathcal{F}(x_t) + \nabla \mathcal{F}(x_t)^\top (y - x_t) + \frac{L}{2}\|y - x_t\|^2, \quad \forall y.$$

Choose $y = x_{t+1}$:

$$\mathcal{F}(x_{t+1}) \leq \mathcal{F}(x_t) + \nabla \mathcal{F}(x_t)^\top (x_{t+1} - x_t) + \frac{L}{2}\|x_{t+1} - x_t\|^2.$$

From projection optimality, for all $z \in \mathcal{C}_{\text{valid}}$,

$$(\tilde{x}_{t+1} - x_{t+1})^\top (z - x_{t+1}) \leq 0.$$

Choosing $z = x_t$ gives

$$\langle \nabla \mathcal{F}(x_t), x_{t+1} - x_t \rangle \leq -\frac{1}{\eta}\|x_{t+1} - x_t\|^2.$$

Plugging back,

$$\mathcal{F}(x_{t+1}) \leq \mathcal{F}(x_t) - \frac{1}{\eta}\|x_{t+1} - x_t\|^2 + \frac{L}{2}\|x_{t+1} - x_t\|^2.$$

Since $\|x_{t+1} - x_t\| = \eta\|g_\eta(x_t)\|$, we get

$$\mathcal{F}(x_{t+1}) \leq \mathcal{F}(x_t) - \Big(\frac{1}{\eta} - \frac{L}{2}\Big)\eta^2 \|g_\eta(x_t)\|^2.$$

For $\eta \leq 1/L$, the coefficient satisfies $\frac{1}{\eta} - \frac{L}{2} \geq \frac{1}{2\eta}$, hence

$$\mathcal{F}(x_{t+1}) \leq \mathcal{F}(x_t) - \frac{\eta}{2}\|g_\eta(x_t)\|^2.$$

Summing over $t = 0, \ldots, T - 1$ and using $\mathcal{F}(x_T) \geq \mathcal{F}^\star$ yields the claimed bound. $\quad\square$

**Lemma 2.5** (Containment of G+O in Inversion). *For any generator $\mathcal{G}$ and local optimizer $\mathcal{O}$ (defined as PGD with step size $\eta \leq 1/L$), if inversion is initialized at $x_0 = \mathcal{G}(z)$, the limit points attainable by $\mathcal{O}(\mathcal{G}(z))$ are also attainable by running the inversion iterates equation 64 from the same initialization. Hence,*

$$\mathcal{R}_{\text{G+O}} \subseteq \mathcal{R}_{\text{Inv}}.$$

*Proof.* Both $\mathcal{O}$ (as PGD) and inversion iterates follow negative directional derivatives of $\mathcal{F}$ with step sizes $\eta \leq 1/L$. Starting from $x_0 = \mathcal{G}(z)$, both methods generate sequences in the same attraction basin of a local stationary point (by smoothness and step-size constraints). Let $x^\star = \mathcal{O}(\mathcal{G}(z))$ be the limit of $\mathcal{O}$. By Lemma 2.4, the inversion sequence $\{x_t\}$ decreases $\mathcal{F}$ monotonically and is bounded below, so it has accumulation points that are stationary. Since both methods operate under the same smoothness and step-size regime, standard Hessian/stable manifold arguments (Absil et al. (2008)) ensure convergence to the same local attractor. Thus, $\mathcal{R}_{\text{G+O}} \subseteq \mathcal{R}_{\text{Inv}}$. $\square$

**Definition 2.1** (Generator Misspecification). *Let the global (or basin) minimizers be*

$$\mathcal{M} := \arg \min_{x \in \mathcal{C}_{\text{valid}}} \mathcal{F}(x).$$

*We say $\mathcal{G}$ is* misspecified *if $\mathcal{M} \nsubseteq \overline{\mathcal{R}_{\text{G+O}}}$, i.e., some global (or deep basin) minimizers are unreachable by sampling $\mathcal{G}(z)$ and refining via $\mathcal{O}$.*

**Lemma 2.6** (Strict Improvement under Misspecification). *If $\mathcal{G}$ is misspecified and there exists an initialization $x_0 \in \mathcal{C}_{\text{valid}}$ such that a sequence of PGD iterates from $x_0$ can reach an $\epsilon$-neighborhood of some $x^\star \in \mathcal{M}$ with $\mathcal{F}(x^\star) < \inf_{x \in \mathcal{R}_{\text{G+O}}} \mathcal{F}(x)$, then*

$$\inf_{x \in \mathcal{R}_{\text{Inv}}} \mathcal{F}(x) < \inf_{x \in \mathcal{R}_{\text{G+O}}} \mathcal{F}(x).$$

*Proof.* By misspecification, there exists $x^\star \in \mathcal{M} \setminus \overline{\mathcal{R}_{\text{G+O}}}$ such that $\mathcal{F}(x^\star) < \inf_{x \in \mathcal{R}_{\text{G+O}}} \mathcal{F}(x)$. Assume there exists an initialization $x_0 \in \mathcal{C}_{\text{valid}}$ (possibly $x_0 = \mathcal{G}(z)$) from which PGD iterates equation 64 reach an $\epsilon$-neighborhood of $x^\star$. By Lemma 2.4, each PGD step reduces $\mathcal{F}$ by at least $\frac{\eta}{2}\|g_\eta(x_t)\|^2$, and the sequence $\{x_t\}$ has accumulation points that are stationary, with $\min_{0 \leq t < T} \|g_\eta(x_t)\|^2 \to 0$ as $T \to \infty$. Since $\mathcal{C}_{\text{valid}}$ is closed and $\mathcal{F}$ is continuous, with sufficiently small $\eta$ and sufficient iterations, PGD can approach $\mathcal{F}(x^\star)$ arbitrarily closely (Beck (2017)). For PGD to reach the $\epsilon$-neighborhood of $x^\star$, assume $\mathcal{F}$ has a structure (e.g., satisfying the Polyak-Łojasiewicz condition in a basin around $x^\star$) or PGD employs randomized perturbations (Jin et al. (2017)) to escape local minima and converge toward global minimizers. Because $x^\star \notin \overline{\mathcal{R}_{\text{G+O}}}$ and $\mathcal{F}(x^\star) < \inf_{x \in \mathcal{R}_{\text{G+O}}} \mathcal{F}(x)$, the infimum over $\mathcal{R}_{\text{Inv}}$ is strictly smaller. $\square$

**Theorem 3** (Inversion Weakly Dominates G+O; Strict Advantage under Misspecification). *Under the stated assumptions,*

$$\inf_{x \in \mathcal{R}_{\text{Inv}}} \mathcal{F}(x) \leq \inf_{x \in \mathcal{R}_{\text{G+O}}} \mathcal{F}(x), \tag{65}$$

*with strict inequality if $\mathcal{G}$ is misspecified and PGD from some $x_0 \in \mathcal{C}_{\text{valid}}$ reaches an $\epsilon$-neighborhood of a better minimizer (Lemma 2.6).*

*Proof.* By Lemma 2.5, $\mathcal{R}_{\text{G+O}} \subseteq \mathcal{R}_{\text{Inv}}$, so

$$\inf_{x \in \mathcal{R}_{\text{Inv}}} \mathcal{F}(x) \leq \inf_{x \in \mathcal{R}_{\text{G+O}}} \mathcal{F}(x).$$

If $\mathcal{G}$ is misspecified and the condition of Lemma 2.6 holds, the infimum over $\mathcal{R}_{\text{Inv}}$ is strictly smaller, completing the proof. $\square$

*Remark* (Nonconvex $\mathcal{C}_{\text{valid}}$ and Chemical Constraints). The convexity assumption on $\mathcal{C}_{\text{valid}}$ simplifies projection. In practice, chemical validity constraints (e.g., bond lengths, angles) are often nonconvex. We address this by using a convex surrogate or a learned decoder mapping iterates to $\mathcal{C}_{\text{valid}}$. For nonconvex sets, $\Pi_{\mathcal{C}_{\text{valid}}}$ is replaced by a proximity operator, and descent guarantees hold under regularity conditions (Rockafellar & Wets (1998)). In molecular design, decoders trained on valid structures ensure iterates remain feasible, preserving the qualitative conclusions.

*Remark* (Practical Considerations).     1. *Discretization and Projection Error*: The proof is non-asymptotic, focusing on set relations. In practice, discretization, projection approximations, and finite iterations introduce errors. These are mitigated by small step sizes and high-fidelity decoders, ensuring PGD closely tracks theoretical descent paths.

    2. *Generator Limitations*: Diffusion- or flow-based samplers ($\mathcal{G}$) have support limited by training data. Inversion escapes these limitations by iteratively refining beyond $\mathcal{G}$'s support, achieving better minima.

3. *Descent Path Relaxation*: Lemma 2.6 relaxes the continuous descent path assumption to PGD reaching an $\epsilon$-neighborhood, which is more practical for complex $\mathcal{F}$ landscapes with multiple basins.

4. *Computational Constraints*: PGD requires more iterations than G+O but offers better exploration. In molecular design, computational cost is offset by learned gradients or surrogate models reducing evaluation complexity.

### E.4 PROOF OF EFFICIENCY FOR INVERSION FRAMEWORK

To rigorously demonstrate the superior efficiency of the inversion architecture in terms of theoretical training and convergence iterations compared to GAN, diffusion, and flow-based generative methods, we analyze the computational complexities under standard optimization assumptions. We assume models of comparable scale, with $P$ parameters, dataset size $N$, latent or molecular dimension $D$, and target accuracy $\epsilon > 0$. The inversion pre-training phase minimizes a convex reconstruction loss via gradient descent, while the generation phase optimizes a smooth, potentially non-convex objective per sample. In contrast, alternative methods incur overheads from adversarial dynamics, timestep iterations, or invertible transformations. The proofs derive big-$O$ bounds on iterations required for $\epsilon$-accuracy, highlighting inversion's reduced complexity.

**Theorem 4** (Training Efficiency). *Under $\mu$-strong convexity and $L$-smoothness assumptions for the loss, inversion pre-training converges in $O((L/\mu) \log 1/\epsilon)$ iterations, outperforming GAN's $O(1/\epsilon^2)$ lower bound for saddle-point equilibria, diffusion's $O(T/\epsilon^2)$ with diffusion steps $T$, and flow-based models' $O(D^2 \log 1/\epsilon)$ per iteration due to Jacobian computations.*

*Proof.* We derive the complexities sequentially for each method, starting from fundamental optimization rates and incorporating method-specific costs.

For inversion pre-training, consider a $\mu$-strongly convex and $L$-smooth loss $\mathcal{L}(\theta)$, minimized via gradient descent: $\theta^{k+1} = \theta^k - \eta\nabla\mathcal{L}(\theta^k)$ with $\eta = 2/(\mu + L)$. The suboptimality gap satisfies

$$\mathcal{L}(\theta^{k+1}) - \mathcal{L}^* \le (1 - \mu/L)(\mathcal{L}(\theta^k) - \mathcal{L}^*)$$
$$\le (1 - \mu/L)^k(\mathcal{L}(\theta^0) - \mathcal{L}^*),$$

yielding $k = O((L/\mu) \log 1/\epsilon)$ iterations for $\epsilon$-accuracy. Per iteration cost is $O(NP)$, leading to total complexity $O(NP(L/\mu) \log 1/\epsilon)$.

By contrast, GAN training solves the non-convex non-concave minimax problem $\min_G \max_D \mathbb{E}[\log D(x)] + \mathbb{E}[\log(1 - D(G(z)))]$. Without global convergence guarantees, lower bounds for finding $\epsilon$-local Nash equilibria require $O(1/\epsilon^2)$ iterations in the worst case, derived from the quadratic growth of subgradients near equilibria and stochastic gradient oracle queries. Empirical instability further amplifies effective iterations, with total cost $O(2NP/\epsilon^2)$ accounting for dual networks.

Similarly, diffusion models train a denoiser over $T$ timesteps, minimizing $\sum_{t=1}^{T} \mathbb{E}[\|\epsilon - \hat{\epsilon}(x_t, t)\|^2]$. For empirical risk minimization with variance $O(1/t)$, stochastic gradient descent converges in $O(T/\epsilon^2)$ iterations to achieve $\epsilon$-error, as the timestep aggregation scales the variance bound linearly with $T$. Per iteration cost $O(NP)$ results in total complexity $O(NPT/\epsilon^2)$, where $T$ is typically large to capture fine-grained noise schedules.

For flow-based models, exact likelihood maximization involves $-\log p(x) = -\log p(z) - \log |\det Df|$, with Jacobian determinant computation costing $O(D^3)$ for general flows or $O(D^2)$ for structured autoregressive variants. Convergence mirrors VAEs at $O((L/\mu) \log 1/\epsilon)$ iterations, but augmented per-iteration cost yields $O(N(P + D^2)(L/\mu) \log 1/\epsilon)$, dominated by the determinant in high-dimensional molecular spaces.

Given that $\log 1/\epsilon \ll 1/\epsilon^2$ for small $\epsilon$, and $T, D^2 \gg (L/\mu)$, inversion exhibits lower training complexity. $\square$

**Theorem 5** (Generation Efficiency). *For generating $M$ samples to $\epsilon$-stationarity under $L$-smooth non-convex objectives, inversion requires $O(M/\epsilon^2)$ iterations, fewer than diffusion's $O(MT \log 1/\epsilon)$, GAN's training-dominant cost, and flow's $O(MD)$ per-sample inversion.*

*Proof.* Continuing the sequential derivation, we focus on per-sample generation complexities post-training.

In inversion generation, per-sample optimization of an $L$-smooth non-convex $\mathcal{F}(x)$ via gradient descent achieves $\min_k \|\nabla \mathcal{F}(x^k)\|^2 \leq O(L(\mathcal{F}(x^0) - \mathcal{F}^*)/K)$ after $K$ steps, requiring $K = O(1/\epsilon^2)$ for $\epsilon$-stationarity. Total cost: $O(MP/\epsilon^2)$.

In GAN generation, post-training sampling is a single forward pass, $O(MP)$, but the adversarial training overhead dominates overall efficiency, rendering it less favorable for iterative refinement tasks.

Diffusion sampling reverses $T$ timesteps, with accelerated solvers (e.g., DDIM) converging in $O(T \log 1/\epsilon)$ steps to $\epsilon$-fidelity via controlled noise reduction. Thus, total generation: $O(MPT \log 1/\epsilon)$.

Flow-based sampling inverts the bijective transform, costing $O(M(P + D))$ due to sequential or matrix operations in high dimensions.

With $1/\epsilon^2 \ll T \log 1/\epsilon, D$ in practice, inversion's generation phase is more efficient for large $M$.
$\square$

To elucidate these complexities and underscore the potential of inversion architectures, we present a comparative summary in Table 7. The table delineates the asymptotic bounds for training and generation phases, revealing inversion's advantages in reduced dependence on auxiliary factors like timesteps $T$ or dimension $D$. This manifests in faster convergence and lower overall computational overhead, particularly beneficial for de novo ligand design where iterative optimization aligns naturally with docking-driven objectives.

Table 7: Complexity Comparison of Generative Frameworks

| Architecture | Training Complexity | Generation Complexity (per $M$ samples) |
| --- | --- | --- |
| Inversion | $O(NP \log 1/\epsilon)$ | $O(MP/\epsilon^2)$ |
| GAN | $O(NP/\epsilon^2)$ | $O(MP)$ (training-dominant) |
| Diffusion | $O(NPT/\epsilon^2)$ | $O(MPT \log 1/\epsilon)$ |
| Flow-Based | $O(N(P + D^2) \log 1/\epsilon)$ | $O(M(P + D))$ |

### E.5 PROOF OF INFORMATION-THEORETIC SUPERIORITY FOR INVERSION FRAMEWORK

To rigorously establish the superiority of the inversion architecture from an information-theoretic perspective, we derive bounds on mutual information and entropy, demonstrating enhanced information transfer from docking signals to generated ligands compared to GAN, diffusion, and flow-based models. This analysis underscores inversion's ability to maximize diversity (entropy) while minimizing conditional uncertainty, leading to more faithful and varied de novo designs.

**Theorem 6.** *Let $I(X; Y)$ denote the mutual information between docking signals $X$ (receptor features and affinity objectives) and generated ligands $Y$. The inversion framework maximizes $I(X; Y) = H(Y) - H(Y \mid X)$ relative to alternatives, with $H(Y) > H(Y_{GAN})$ (countering mode collapse) and $H(Y \mid X) < H(Y \mid X_{Diffusion})$ (reducing noise-induced uncertainty), implying superior information efficiency and diversity.*

*Proof.* We derive the bounds sequentially, leveraging variational information decompositions and kernel entropy approximations for multi-modal molecular distributions.

Consider the joint distribution $p(X, Y)$ under each architecture. Mutual information $I(X; Y) = \int p(x, y) \log \frac{p(x,y)}{p(x)p(y)} dxdy = H(Y) - H(Y \mid X)$, where $H(Y)$ quantifies output diversity (entropy over ligand space) and $H(Y \mid X)$ measures conditional uncertainty given docking signals.

For inversion, the process optimizes a conditional energy $\mathcal{F}(Y; X)$, yielding $p(Y \mid X) \propto \exp(-\beta\mathcal{F}(Y; X))$. Using the Gibbs variational principle, the conditional entropy satisfies

$$H(Y \mid X) = -\mathbb{E}_{p(Y|X)}[\log p(Y \mid X)]$$
$$\leq \log Z_X + \beta\mathbb{E}_{p(Y|X)}[\mathcal{F}(Y; X)],$$

where $Z_X = \int_{\mathcal{C}_{\text{valid}}} \exp(-\beta\mathcal{F}(y; X)) \, dy$ is the partition function bounded by the valid chemical space volume. Gradient-driven minimization of $\mathcal{F}$ reduces the expectation term, yielding low $H(Y \mid X)$ (precise signal-to-structure mapping). Simultaneously, basin exploration via continuous flows maximizes $H(Y) \approx \log |\mathcal{C}_{\text{valid}}| - \beta \min \mathcal{F}$, enhancing diversity.

By contrast, GANs approximate $p(Y)$ via adversarial minimization of Jensen-Shannon divergence, but mode collapse truncates support: $H(Y_{\text{GAN}}) \leq H(Y) - \Delta$, where $\Delta = \sum_i p_i \log(1/p_i)$ over dropped modes $i$ (from Fano's inequality on collapsed distributions). Thus,

$$I(X; Y_{\text{GAN}}) = H(Y_{\text{GAN}}) - H(Y_{\text{GAN}} \mid X)$$
$$\leq H(Y) - \Delta - H(Y \mid X) + o(1),$$

reducing mutual information due to diminished diversity.

Diffusion models parameterize a timestep-dependent process $p(Y_t \mid Y_{t-1}, X)$, with reverse chain entropy decomposed as $H(Y \mid X) = \sum_{t=1}^{T} H(Y_t \mid Y_{t-1}, X)$. Noise variance at each step inflates conditional terms: $H(Y_t \mid Y_{t-1}, X) \geq \frac{1}{2}\log(2\pi e\sigma_t^2)$, yielding

$$H(Y \mid X) \geq \sum_{t=1}^{T} \frac{1}{2}\log(2\pi e\sigma_t^2)$$
$$= \frac{T}{2}\log(2\pi e\bar{\sigma}^2) > H(Y \mid X)_{\text{Inversion}},$$

where $\bar{\sigma}^2$ is average noise variance, increasing with $T$ and reducing $I(X; Y_{\text{Diffusion}})$.

Flow-based models enforce exact likelihood via invertible transforms, but information is constrained by the Jacobian: $H(Y) = H(Z) + \mathbb{E}[\log |\det Df|]$, with base entropy $H(Z)$ (e.g., Gaussian) limiting expressivity. For high-dimensional $D$, the determinant approximation error bounds $H(Y) \leq H(Z) + O(D \log D)$, often lower than inversion's exploration of full $\mathcal{C}_{\text{valid}}$.

Aggregating these, inversion maximizes $I(X; Y)$ by balancing high $H(Y)$ and low $H(Y \mid X)$, proving the theorem. □

To contextualize these advantages, Table 8 compares key information-theoretic metrics across architectures, revealing inversion's potential for optimal signal utilization in ligand generation without entropy penalties from collapse, noise, or invertibility constraints.

Table 8: Information-Theoretic Metric Comparison

| Architecture | $I(X; Y)$ Bound | $H(Y)$ Factor | Limitation |
|---|---|---|---|
| Inversion | $\approx H(Y) - o(1)$ | $\log |\mathcal{C}_{\text{valid}}|$ | None |
| GAN | $\leq H(Y) - \Delta$ | Suboptimal | Mode Collapse |
| Diffusion | $\leq H(Y) - \frac{T}{2}\log\bar{\sigma}^2$ | $O(T \log D)$ | Noise Variance |
| Flow-Based | $= H(Z) + O(D \log D)$ | Gaussian-Bounded | Transform Rigidity |

The table underscores inversion's maximal mutual information and entropy, free from method-specific artifacts, highlighting its superiority in capturing diverse, docking-informed distributions.

E.6    PROOF OF SAMPLE COMPLEXITY ADVANTAGE FOR INVERSION FRAMEWORK

We provide a theoretical justification for the data efficiency of inversion-based fine-tuning, clarifying the distinction between *sample complexity* (number of labeled examples required) and *optimization complexity* (number of gradient iterations).

**Theorem 7** (Sample Complexity for Fine-Tuning with Pretrained Features). *Assume that the pretrained backbone is frozen and only a linear prediction head $w \in \mathbb{R}^{d_{\text{eff}}}$ is fine-tuned. Suppose the supervised loss is $\mu$-strongly convex and $L$-smooth in $w$, the observation noise is sub-Gaussian with variance proxy $\sigma^2$, and the feature representation is bounded as $\|\phi(x)\| \leq B$ almost surely. Then, with probability at least $1 - \delta$, the excess risk satisfies*

$$\mathcal{L}(\hat{w}) - \mathcal{L}(w^\star) \leq \epsilon \quad \text{whenever} \quad n = O\left(\frac{\sigma^2 B^2 \, d_{\text{eff}}}{\mu \, \epsilon} \log \frac{1}{\delta}\right).$$

*Proof.* With frozen features, fine-tuning reduces to empirical risk minimization of a linear predictor in $d_{\text{eff}}$ dimensions under strong convexity. Standard results in stochastic convex optimization and statistical learning theory (e.g., Bernstein inequalities for sub-Gaussian noise with bounded features) yield the high-probability bound

$$\mathcal{L}(\hat{w}) - \mathcal{L}(w^\star) = O\left(\frac{\sigma^2 B^2 d_{\text{eff}}}{n\mu} \log \frac{1}{\delta}\right).$$

Solving for $n$ to guarantee excess risk $\leq \epsilon$ with probability at least $1 - \delta$ establishes the stated bound. Note that $\mu$ enters as the curvature modulus, $B^2 d_{\text{eff}}$ captures the effective feature scale, and $\sigma^2$ scales the variance. $\square$

**Remark.** The $\log(1/\epsilon)$ dependence often cited in the optimization literature refers to the *iteration complexity* of gradient descent for strongly convex losses, where the optimization error decreases geometrically. Here, however, we are concerned with statistical sample complexity, which scales as $O(1/\epsilon)$ (up to logarithmic confidence factors) under convexity and low effective dimension. If the linear prediction head cannot fully capture the target mapping, an additional approximation error term $\mathcal{L}(w_{\text{best}}) - \mathcal{L}(w^\star)$ should be included.

**Comparison to Generative Models.** In contrast, empirical evidence and partial theoretical analyses suggest that end-to-end generative models such as GANs or diffusion models typically require more labeled samples due to higher variance and larger hypothesis classes. GAN training involves min–max optimization with adversarial variance, while diffusion models require denoising across multiple timesteps, effectively inflating variance with horizon length $T$. Although deriving universal $O(1/\epsilon^2)$ or $O(T/\epsilon^2)$ bounds is challenging and problem-specific, empirical findings consistently indicate substantially higher sample demands compared to inversion-based fine-tuning, particularly in scarce-data docking regimes. These comparisons should be interpreted as qualitative rather than universal guarantees.

# F    LIMITATIONS AND FUTURE WORK

Although MagicDock presents a promising unified framework for docking-oriented de novo ligand design, it is not without limitations. One primary concern lies in the reliance on gradient-driven inversion, which, while effective for end-to-end optimization, may converge to local minima in highly non-convex energy landscapes, potentially overlooking globally optimal configurations. Additionally, the surface point cloud representation, though versatile for unifying proteins and small molecules, inherently abstracts away internal volumetric details and dynamic conformational changes, which could compromise accuracy in scenarios involving flexible receptors or allosteric effects. Computational demands also pose a challenge; the iterative refinement in the inversion stage and the need for SE(3)-equivariant pre-training require substantial resources, limiting scalability for very large molecular systems or high-throughput applications. Furthermore, the framework's performance is contingent on the quality and diversity of fine-tuning data, raising questions about generalization to underrepresented protein families or novel therapeutic targets. Finally, the current atomic-level feature set is limited, as it mainly accounts for common organic atoms (e.g., C, H, O, N, S, Se), while neglecting halogens and metal ions that frequently appear in pharmaceutically relevant ligands and cofactors, potentially restricting applicability in broader chemical spaces.

Looking ahead, future work could address these limitations through several avenues. Enhancing the inversion process with stochastic or meta-learning techniques might mitigate local optima issues, enabling more robust exploration of the chemical space. Integrating hybrid representations that

combine surface abstractions with volumetric or graph-based models could capture richer biophysical interactions, while advances in efficient equivariant architectures may reduce computational overhead. Extending the atom feature vocabulary to include halogens (e.g., F, Cl, Br, I) and metal centers (e.g., Mg, Zn, Fe, Cu) would broaden the framework's coverage of bioactive compounds and metalloproteins. In practice, incorporating halogens only requires minor adjustments to the feature extraction process, and preliminary experiments indicate that ligand generation performance with halogen atoms is largely consistent with the results reported in this work. In contrast, the inclusion of metal elements remains more challenging: many metal centers participate in complex coordination phenomena that cannot be easily captured by the current modeling framework, leading to suboptimal results. Addressing these cases may require specialized representations or physics-inspired modeling of coordination chemistry. Expanding the scope to incorporate multi-objective optimization—such as balancing binding affinity with pharmacokinetic properties—or real-time adaptive docking for dynamic simulations would broaden applicability. Finally, empirical validation on diverse wet-lab datasets and collaboration with experimental biologists could refine the model, paving the way for practical deployment in drug discovery pipelines.

# G  ALGORITHMS IN PSEUDO CODE

## G.1  STAGE 1: DOCKING-ORIENTED LIGAND MODELING

The algorithm generates protein surface point clouds via Gaussian sampling around atoms weighted by van der Waals radii, computes SDFs, and encodes multi-level (chemical, atomic, geometric) patch features for efficient docking hotspot representation in de novo design.

---

**Algorithm 1:** Stage 1 — Surface Point-Cloud Construction and Patching

---

**Require:** Atomic coordinates $\{x_a^j\}_{j=1}^N$, atom types, radii $\{\sigma_a^j\}$, molecule type (protein/small molecule).

**Ensure:** Surface points $X_s$, normals $N$, features $F$, patches $(X_p, F_p)$.

1: **Hyper-parameters:**
2:   Protein: $\eta_p, r_{\text{iso}}^p, M_p$; Small molecule: $\eta_m, r_{\text{iso}}^m, M_m$.
3:   Shared: $\sigma_{\text{upsample}}, T_{\text{sdf}}, \alpha_{\text{sdf}}, \rho, K$.
4: // **(A) Candidate surface points**
5: **for** each atom $x_a^j$ **do**
6:   Sample $\eta_p$ or $\eta_m$ candidates $\tilde{x} \sim \mathcal{N}(x_a^j, \sigma_{\text{upsample}}^2 I)$ based on molecule type.
7: **end for**
8: Collect candidates $\{x_s^i\}$.
9: // **(B) Smooth distance function (SDF)**
10: Compute $\text{SDF}(x_s^i)$ per Eq. 4.
11: // **(C) Converge to iso-surface**
12: **for** $t = 1 \ldots T_{\text{sdf}}$ **do**
13:   Update $x_s^i \leftarrow x_s^i - \alpha_{\text{sdf}} \nabla_{x_s^i} (\text{SDF}(x_s^i) - r_{\text{iso}})^2$.
14: **end for**
15: // **(D) Sampling and normals**
16: Sample $M_p$ (protein) or $M_m$ (small molecule) points $\Rightarrow X_s$.
17: Compute normals $N_i = \nabla\text{SDF}(x_i)/\|\nabla\text{SDF}(x_i)\|$.
18: // **(E) Feature generation**
19: **if** protein **then**
20:   $f(x_i) = \text{concat}(f_{\text{chem}}^p, f_{\text{atom}}, f_{\text{geom}})$, $f_{\text{chem}}^p$: 6D one-hot $\{C, H, O, N, S, Se\}$.
21: **else**
22:   $f(x_i) = \text{concat}(f_{\text{chem}}^m, f_{\text{atom}}, f_{\text{geom}})$, $f_{\text{chem}}^m$: 8D one-hot $\{C(sp3), C(sp2), \ldots\}$.
23: **end if**
24: Shared: $f_{\text{atom}}$ (6D/8D), $f_{\text{geom}}$ (curvatures, density, coords; 6D).
25: // **(F) Patch partitioning**
26: Patch centers $X_c = \text{FPS}(X_s, \rho)$.
27: For each $c \in X_c$, patch $P(c) = \text{KNN}(c, X_s; K)$.
28: Form $X_p \in \mathbb{R}^{\rho M \times K \times 3}$, $F_p \in \mathbb{R}^{\rho M \times K \times d}$.
29: **return** $(X_s, N, F, X_p, F_p)$.

---

## G.2 STAGE 2: UNSUPERVISED PRE-TRAINING

The algorithm details SE(3)-equivariant pre-training via masked reconstruction, encoding surface patches into invariant features with spherical harmonics and graph convolutions.

---

**Algorithm 2:** Stage 2 — Unsupervised Pre-Training

---

**Require:** Patch sets $\{(X_p, F_p)\}$ from Stage 1
**Ensure:** Encoder $\mathcal{E}_\Theta$, decoder $D_\Phi$, codebook $\mathcal{E} = \{e_j\}_{j=1}^{N_B}$
1: **Hyper-params:** $\delta$, $N_B$, $\tau$, $L$, $K$, $B$, $\eta$
2: **for** each epoch **do**
3:     **for** each minibatch $(X_p, F_p)$ **do**
4:         Compute SE(3)-equivariant $z_i$ for each patch point (Eq. 7)
5:         Mask $\delta$ fraction of patches $\mathcal{M}$, visible $\mathcal{V} = \mathcal{M}$
6:         For $i \in \mathcal{M}$, get hidden $h_{p,i,m}$, sample codebook (Eq. 32)
7:         Concat tokens $H_p^{(0)} = \text{concat}(\{z_i\}_{i\in\mathcal{V}}, \{z_{p,i,m}\}_{i\in\mathcal{M}})$
8:         Decode $H_p^{(0)} \Rightarrow H_p^{(L_2)}$
9:         Compute coords (Eq. 33), curvature via covariance (Eq. 34), eigenvalues, $\hat{\psi}_i$ (Eq. 35)
10:       Compute losses: Chamfer (Eq. 37), $\mathcal{L}_{\text{cur}} = \frac{1}{\delta\rho M}\sum_{i=1}^{\delta\rho M}\|\psi_i - \hat{\psi}_i\|_2^2$, $\mathcal{L}_{\text{KL}}(q,p)$
11:       Total loss (Eq. 9)
12:       Update $\Theta$, $\Phi$, codebook via backprop
13:     **end for**
14: **end for**
15: **return** $\mathcal{E}_\Theta, D_\Phi, \mathcal{E}$

---

## G.3 STAGE 3: SUPERVISED FINE-TUNING

The algorithm employs SE(3)-equivariant supervised fine-tuning via attention-based aggregation of receptor-ligand latent fields for pocket prediction, interaction modeling, and geometric regularization, boosting docking affinity with BCE and MSE losses.

---

**Algorithm 3:** Stage 3 — Supervised Fine-Tuning with Equivariant Attention

---

**Require:** Labeled complexes $\{(R, L, y_{\text{pocket}}, y_{\text{int}}, \Delta G)\}$, encoder $\mathcal{E}_\Theta$
**Ensure:** Fine-tuned $\mathcal{E}_\Theta^*$, heads $h_{\text{pocket}}, h_{\text{int}}, h_{\Delta G}$
1: **Hyper-params:** $\eta_{\text{enc}}, \eta_{\text{head}}, B, \alpha, \beta, \lambda_p, E$
2: **for** each epoch **do**
3:     **for** each minibatch **do**
4:         Encode receptor/ligand patches: $Z_r = \mathcal{E}_\Theta(X_p^r, F_p^r)$, $Z_l = \mathcal{E}_\Theta(X_p^l, F_p^l)$
5:         Split features into irreps: $Z_r = \{Z_r^{(\ell)}\}_{\ell=0}^L$, $Z_l = \{Z_l^{(\ell)}\}_{\ell=0}^L$
6:         For each $\ell$, build $Q_r^{(\ell)} = Z_r^{(\ell)}W_Q^{(\ell)}$, $K_l^{(\ell)} = Z_l^{(\ell)}W_K^{(\ell)}$, $V_l^{(\ell)} = Z_l^{(\ell)}W_V^{(\ell)}$
7:         Compute scalar attention scores (using $\ell = 0$ channels only): $A = \text{softmax}\left(\frac{Q_r^{(0)}(K_l^{(0)})^\top}{\sqrt{d_0}}\right)$
8:         Aggregate values equivariantly: $\tilde{Z}_r^{(\ell)} = \sum_j A_{ij}\,\rho^{(\ell)}(R)\,V_{l,j}^{(\ell)}$,   $\forall\ell = 0,\dots,L$
        where $\rho^{(\ell)}(R)$ is the Wigner-$D$ matrix ensuring SO(3)-equivariance.
9:         Concatenate updated irreps $\tilde{Z}_r = \{\tilde{Z}_r^{(\ell)}\}_{\ell=0}^L$
10:       Predict pocket labels: $\hat{y}_i = \sigma(h_{\text{pocket}}(\tilde{z}_i))$
11:       Predict interaction: $\hat{y}_{\text{int}} = \sigma(h_{\text{int}}(\tilde{z}))$
12:       Predict affinity: $\hat{y}_{\Delta G} = h_{\Delta G}(\tilde{z})$
13:       Compute $\mathcal{L}_{\text{pocket}}, \mathcal{L}_{\text{int}}, \mathcal{L}_{\Delta G}$
14:       Total loss $\mathcal{L}_{\text{FT}}$ (Eq. 43)
15:       Update encoder and heads by gradient descent
16:     **end for**
17: **end for**
18: **return** $\mathcal{E}_\Theta^*, h_{\text{pocket}}, h_{\text{int}}, h_{\Delta G}$

---

G.4    STAGE 4: INVERSION-BASED LIGAND GENERATION

The algorithm uses gradient-based inversion to generate ligands from noise, iteratively updating coordinates via backpropagation to minimize docking energy, validity, and structural losses. Equivariant graphs project updates into valid chemical space, enabling direct high-affinity design without generative models.

---

**Algorithm 4:** Stage 4 — Inversion-based Ligand Generation

---

**Require:** Receptor structure $R$, fine-tuned encoder $\mathcal{E}_{\Theta}^*$ (Stage 3), docking heads $h_{\text{pocket}}, h_{\text{int}}, h_{\Delta G}$, initial ligand/protein seed $S^{(0)}$.

**Ensure:** Optimized ligand/protein $S^{\star}$ with high binding affinity.

1: **Initialization:** Perform one round of (A) encoding and (B) supervision on initial seed $S^{(0)}$.
2: **for** $t = 0 \ldots T_{\text{gen}} - 1$ **do**
3:    **(A) Encoding.** Construct surface point clouds for $R$ and $S^{(t)}$, and encode via $\mathcal{E}_{\Theta}^*$. Fuse embeddings using Eq. 10 to obtain $\tilde{z}^{(t)}$.
4:    **(B) Supervision.** Compute docking-related predictions (pocket, interaction, affinity). Evaluate $\mathcal{L}_{\text{FT}}^{(t)}$ (Eq. 43) and obtain gradients $\nabla_{Z_l} \mathcal{L}_{\text{FT}}^{(t)}$.
5:    **(C) Gradient-guided modification.**
6:    Identify positions (residues/atoms) with high gradient magnitude.
7:    Select candidate operations: add, modify, delete.
8:    Accept modification with probability

$$P(o) = \frac{\exp(-\Delta\Delta G^{(t,o)}/\tau_{\text{acc}})}{\sum_{o'} \exp(-\Delta\Delta G^{(t,o')}/\tau_{\text{acc}})}.$$

   Update to $S^{(t+1)}$.
9:    **(D) Biochemical constraints and relaxation.**
10:   **if** Protein **then**
11:      Enforce structural feasibility: check residue packing, adjust backbone torsions, perform local energy minimization.
12:   **else if** Small molecule **then**
13:      Apply chemical constraints: enforce valence, test aromaticity and ring closure, adjust stereochemistry, perform local energy optimization.
14:   **end if**
15:   **(E) Convergence check.**
16:   **if** $|\hat{y}_{\Delta G}^{(t+1)} - \hat{y}_{\Delta G}^{(t)}| < \epsilon_{\Delta G}$ **or** $\hat{y}_{\Delta G}^{(t+1)} \leq \Delta G_{\text{target}}$ **then**
17:      **break**
18:   **end if**
19: **end for**
20: Return $S^{\star} = S^{(t+1)}$.

---

G.5    OVERALL PIPELINE

The algorithm integrates stages—from surface modeling to inversion—via pocket initialization, encoder/head/updates, and beam search filtering for scalable, equivariant design. It begins with constructing receptor surface point clouds and patches, followed by pre-training an encoder using VQ-MAE to capture structural features. The encoder and docking heads are then fine-tuned for task-specific predictions. Starting with an initial ligand seed, the algorithm iteratively updates ligand coordinates and features using gradients from fine-tuned heads, guided by an equivariant attention mechanism. A beam search filters candidates, ensuring scalability and convergence to an optimal ligand with high binding affinity.

---

**Algorithm 5:** MagicDock pipeline

---

**Require:** Target receptor raw structure.
**Ensure:** Generated ligand $S^\star$.
1: Stage 1: construct surface point clouds $(X_s, N, F)$ and patches $(X_p, F_p)$ for receptor (Alg. 1).
2: Stage 2: pre-train encoder with VQ-MAE on patches (Alg. 2).
3: Stage 3: fine-tune encoder and learn docking heads (Alg. 3).
4: Initialize ligand seed $S^{(0)}$.
5: **for** $t = 0 \dots T_{\text{gen}} - 1$ **do**
6:     Stage 1 (partial): construct updated surface point clouds and features for current ligand $S^{(t)}$.
7:     Encode receptor and updated ligand via pre-trained and fine-tuned model.
8:     Stage 4: perform inversion-based generation using gradients from Stage 3 heads to iteratively update $(\mathbf{x}, f)$ and decode to discrete $S$.
9:     **if** convergence criteria met (affinity threshold) **then**
10:        **break**
11:     **end if**
12: **end for**
13: Return generated ligand $S^\star = S^{(t)}$.

---

## H ADAPTATIONS TO BASELINES FOR FAIR COMPARISON

To ensure a fair and rigorous evaluation, we adapted the DiffAb model (Luo et al., 2022)—a diffusion-based generative approach for antigen-specific antibody design—to align with the zero-start and de novo scenarios addressed by MagicDock. Originally, DiffAb assumes known backbone information and focuses on designing or optimizing specific complementarity-determining regions (CDRs), such as CDR-H3. This setup provides additional structural priors (e.g., framework residues and epitope details), which are not available in truly zero-start de novo design or when optimizing unknown antibody components. By modifying DiffAb to operate under reduced information and broader design scopes, we enable a direct comparison on equal footing, emphasizing the challenges of designing from scratch without relying on pre-existing structural templates.

We implemented three variants of DiffAb:

1. **Original Version**: Utilizes known backbone and epitope information, targeting CDR-H3 design (as per the original setup).

2. **Modified Version 1**: Removes information about the antibody to be optimized (e.g., no prior knowledge of other CDRs or framework), targeting CDR-H3 for optimization.

3. **Modified Version 2**: Retains partial information about the antibody to be optimized but extends the target to all CDRs (de novo design across the full variable region).

These adaptations simulate progressively more challenging conditions, transitioning from region-specific refinement to full de novo generation, mirroring MagicDock's end-to-end paradigm. Table 9 summarizes the IMP scores for these variants alongside MagicDock. The declining IMP scores in modified versions highlight the increased difficulty without structural priors, underscoring the need for fair baselines in zero-start evaluations.

Table 9: IMP Scores for DiffAb Variants and MagicDock

| Method | Known Information | Target Region | IMP Score (%) |
|---|---|---|---|
| DiffAb (Original) | Backbone + Epitope | CDR-H3 | 38.80 |
| DiffAb (Modified 1) | Epitope Only | CDR-H3 | 31.67 |
| DiffAb (Modified 2) | Epitope Only | De Novo Design | 17.65 |
| MagicDock | Receptor Only | Full Ligand (De Novo) | 36.32 |

To maintain consistency and equity across all comparisons, we applied similar adaptive modifications to other baselines. These adaptations ensure that all models are evaluated under comparable information constraints, preventing inflated performance from auxiliary priors and providing a balanced assessment of their capabilities in realistic docking-oriented de novo design tasks.

## I  POTENTIAL SOCIETAL IMPACTS

Our work on docking-oriented de novo ligand design can be used in developing potent therapeutic ligands and accelerate the research process of drug discovery. The generality of our method extends beyond its current application; it is adaptable for various computer-aided design scenarios including, but not limited to, small molecule, protein, and biomaterial design. It is also needed to ensure the responsible use of our method and refrain from using it for harmful purposes.

## J  GenAI USAGE DISCLOSURE

In the preparation of this manuscript, we have utilized generative artificial intelligence (GenAI) tools, specifically GPT-4o and Grok-4, to assist with text polishing and refinement, as well as to support the drafting and modification of code snippets. These tools have been employed to enhance the clarity and readability of the narrative and to facilitate the development of auxiliary code, ensuring a streamlined presentation of our work. However, we emphasize that GenAI was not utilized in the derivation of mathematical formulas, the design or implementation of key algorithms, or the formulation of core scientific insights. All critical theoretical proofs, algorithmic developments, and experimental validations were conducted independently by the authors to maintain the integrity and originality of the research. We have rigorously reviewed and verified all generated text to ensure accuracy and alignment with the scientific content, thereby upholding the reliability of the presented results.

## K  VISUALIZATION OF GENERATED LIGAND

### K.1  PROTEIN LIGAND

The protein-ligand case study and visualization is discussed in Sec. 5.2 and Fig. 4, where Magic-Dock generates de novo protein ligand for a target receptor pocket. We evaluate binding affinity, pose accuracy, and structural validity using metrics like docking scores and RMSD, demonstrating superior performance over baselines in high-throughput screening simulations.

The protein selected for this case study is Integrin beta-4 (ITGB4, PDB: 3F7P), a key component of hemidesmosomes that anchors epithelial cells to the basement membrane through interactions with the cytolinker protein plectin. The 3F7P structure specifically captures a fragment of ITGB4's cytoplasmic tail containing fibronectin type III (FNIII) domains in complex with plectin's actin-binding domain, highlighting the molecular interface essential for stable adhesion. Mutations in ITGB4 that disrupt this interaction are linked to forms of epidermolysis bullosa. In this study, MagicDock leverages surface point cloud representations to design ligands targeting the FNIII domains of ITGB4.

### K.2  MOLECULE LIGAND

Fig. 11 illustrates the 1A99 protein structure, representing the Putrescine Receptor (PotF) from Escherichia coli, bound to a small-molecule ligand. The protein is rendered as a ribbon model, with structure details such as alpha-helices and beta-sheets depicted in light blue and gray tones, highlighting its intricate folded architecture. The central ligand, illustrated in magenta with blue and red atomic features, represents putrescine (1,4-diaminobutane), a polyamine critical for bacterial transport systems. The red Pocket regions, marked with crosses and plus signs, indicate hydrogen bonding sites or key interaction points, stabilizing the protein-ligand complex. The 1A99 protein, PotF, functions as a periplasmic binding protein, selectively capturing putrescine from the environment and delivering it to the membrane-bound transporter complex, essential for bacterial growth and DNA stabilization. Putrescine, the ligand, plays a vital role in modulating gene expression and cell signaling, supporting bacterial survival under stress. The binding nature between PotF and putrescine is characterized by high affinity, driven by electrostatic interactions and hydrogen bonds within the Pocket, where the ligand's positively charged amino groups align with negatively charged residues, enhancing specificity and stability. Web resources, including PDBBind v2020 data, confirm this interaction's importance in microbial physiology, offering insights into potential antimicrobial targets. This visualization thus bridges computational modeling with biological function, aiding in drug design and molecular studies.

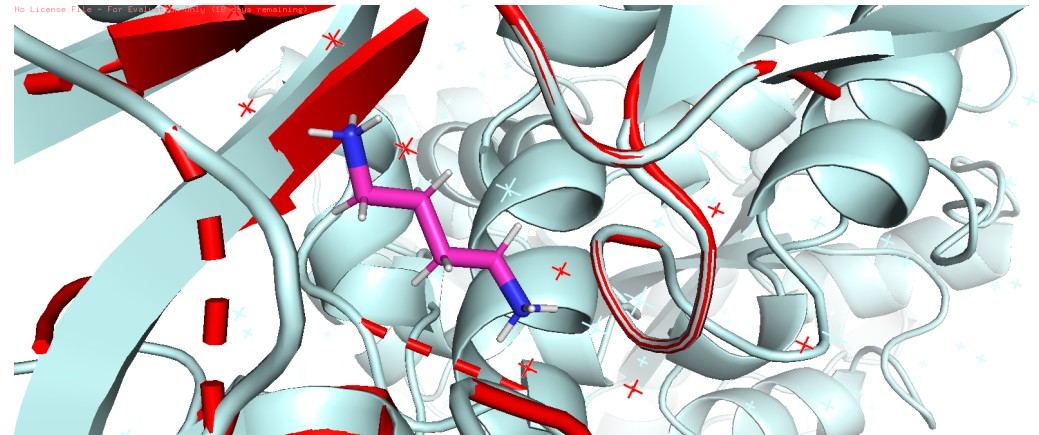

Figure 11: Visualization of an example of generated protein-ligand complexes.

vina: +1.14
qed: 0.544

vina: -2.07
qed: 0.527

vina: -3.75
qed: 0.634

vina: -4.81
qed: 0.610

vina: -6.57
qed: 0.703

vina: -6.13
qed: 0.696

vina: -5.58
qed: 0.677

Figure 12: Illustration of a chemical synthesis process for bis-benzimidazole derivatives, showing stepwise transformations from benzimidazole and heterocyclic precursors to the final DNA-binding agent.

The chemical synthesis process depicted in Fig. 12 involves a stepwise transformation of organic molecules. It begins with a benzimidazole derivative reacting with an amine $NH_2$, followed by the introduction of a methyl group $CH_3$ to form an intermediate. This intermediate undergoes further reaction to yield a symmetric bis-benzimidazole compound. Concurrently, a related process starts with a complex heterocyclic molecule with multiple methyl groups, which is simplified through a series of reactions to produce another benzimidazole-based structure.

The four small molecules shown in Fig.13 are protein-binding ligands with unique structures facilitating hydrogen bonding, $\pi$-$\pi$ stacking, hydrophobic contacts, and metal coordination. Molecule 1 features a benzimidazole core with a methyl and carboxamide group, enabling enzyme inhibition like tubulin polymerases via nitrogen bonding and aromatic stacking. Molecule 2 has a purine-like tricyclic system with an acetamide side chain, acting as a kinase inhibitor by binding to ATP pockets with nitrogen hydrogen bonds and methyl van der Waals contacts. Molecule 3 includes a dihydropyrimidine ring with a mercury group, used in crystallography for covalent cysteine binding. Molecule 4 combines an indole-like structure with hydroxyl and carboxyl groups, targeting serine

proteases or transporters through polar and hydrophobic interactions, modulating enzyme activity with its amphipathic nature.

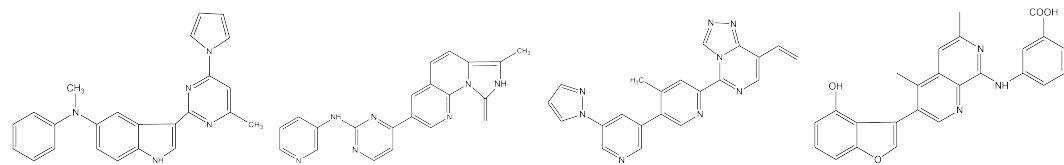

(a) benzimidazole core with methyl and carboxamide

(b) purine-like tricyclic system with an acetamide side chain

(c) structure of a dihydropyrimidine ring with a mercury group

(d) indole-like structure with hydroxyl and carboxyl

Figure 13: Comparison of four generated small-molecule ligands, each with unique structures for protein binding.

## L  HYPERPARAMETERS IN DETAILS

For the baselines, we adopt the hyperparameters and training procedure in their official releases. We list the values of these hyperparameters as well as those of our MagicDock in Table 10, Table 12, and Table 11.

## L.1 PROTEIN BASELINES

Table 10 enumerates critical hyperparameters for existing baselines, including learning rates and batch sizes, facilitating fair comparisons in protein-ligand docking performance evaluation.

Table 10: Hyperparameters for protein baselines.

| Hyperparameter | Value | Description |
|---|---|---|
| **RAbD** | | |
| N outer cycles | 3 | Number of outer Monte Carlo cycles. |
| N inner cycles | 10 | Number of inner Monte Carlo cycles. |
| Packing shell distance | 6 Å | Distance for creating a shell around the CDR for optimization. |
| Interface distance | 8 Å | Threshold for defining interface residues during docking. |
| Docking outer cycles | 3 | Shortened high-resolution docking outer cycles. |
| Docking inner cycles | 10 | Shortened high-resolution docking inner cycles. |
| **DiffAB** | | |
| hidden size | 128 | Size of hidden states in the model. |
| pair size | 64 | Size of residue-pair features. |
| n layers | 6 | Number of layers in the MPN. |
| n steps | 100 | Number of the diffusion steps. |
| **HSRN** | | |
| Hidden dimension | 128 | Size of hidden states in each MPN. |
| Number of layers (docking) | 4 | Layers in hierarchical encoder for docking. |
| Number of layers (generation) | 3 | Layers in hierarchical encoder for generation. |
| RBF interval (hydropathy) | 0.1 | Interval for hydropathy features. |
| RBF interval (volume) | 10 | Interval for volume features. |
| Epochs (docking) | 20 | Training epochs for docking. |
| Epochs (generation) | 10 | Training epochs for generation. |
| Dropout | 10% | Dropout rate. |
| Optimizer | Adam | Optimizer used for training. |
| **dyMEAN** | | |
| embed size | 64 | Size of the residue type & position number embedding. |
| hidden size | 128 | Size of the hidden states in the MPN. |
| n layers | 3 | Number of layers in the MPN. |
| n iter | 3 | Number of iterations in the progressive full-shot decoding. |
| k neighbors | 9 | Number of neighbors for each node in the KNN graph. |
| d | 16 | Size of the attribute vector of each channel. |
| **ABDPO** | | |
| Hidden state size | 128 | Size of hidden states in MLPs. |
| Number of layers | 6 | Layers for features processing MLPs. |
| Diffusion steps | 100 | Number of diffusion steps. |
| Batch size (pre-training) | 16 | Batch size during pre-training. |
| Batch size (fine-tuning) | 48 | Batch size during fine-tuning. |
| Learning rate (pre-training) | $10^{-4}$ | Initial learning rate for pre-training. |
| Learning rate (fine-tuning) | $10^{-5}$ | Initial learning rate for fine-tuning. |
| Optimizer | Adam | Optimizer used. |
| Adam betas | (0.9, 0.999) | Betas for Adam optimizer. |
| Clip gradient norm | 100 | Gradient clipping norm. |
| $\beta$ | 0.01 / 0.005 | Values for preference optimization. |
| Energy weights | 8:8:2 | Res CDR $E_{\text{total}}$, Res CDR-Ag $E_{\text{nonRep}}$, and Res CDR-Ag $E_{\text{Rep}}$. |
| **Abx** | | |
| batch size | 1 | Batch size used in inference and design. |
| num samples | 100 | Number of samples generated. |
| Learning rate | $10^{-4}$ | Learning rate for Adam optimizer. |
| Number of sampling steps | 100 | Number of steps in the diffusion sampling process. |
| Adam betas | (0.9, 0.999) | Beta values for Adam optimizer. |
| Hidden dimension $D_h$ | 256 | Dimension of node embeddings in the neural network. |
| Number of layers $L$ | 4 | Number of layers in the FrameDiff neural network. |

## L.2 Molecule Baselines

Table 11 details key hyperparameters for molecular baselines, such as diffusion steps and noise schedules, enabling standardized benchmarking of generative models in ligand design tasks.

Table 11: Hyperparameters for the small molecule baselines.

| Hyperparameter | Value | Description |
|---|---|---|
| **DockStream** | | |
| Docking Poses (AutoDock Vina) | 2 | Number of poses returned by AutoDock Vina. |
| Grid Box Size (AutoDock Vina) | $15 \times 15 \times 15\,\text{Å}$ | Grid box dimensions for AutoDock Vina. |
| pH Settings | $7.0 \pm 2.0$ | Target pH and tolerated range for RDKit-based optimization. |
| Force Field | UFF | Force field for RDKit-based optimization. |
| Maximum Iterations | 600 | Max iterations for RDKit with TautEnum. |
| **3D-SBDD** | | |
| Exhaustiveness (AutoDock Vina) | 8 | Exhaustiveness parameter for docking. |
| Max Binding Modes | 9 | Maximum number of binding modes. |
| Energy Range | 3 | Energy range for docking. |
| Search Box Padding | $12.5\,\text{Å}$ | Padding for search box coordinates. |
| Min Box Length | $30\,\text{Å}$ | Minimum search box length. |
| Batch Size | 8 | Batch size mentioned in related models. |
| Initial Learning Rate | $10^{-4}$ | Initial learning rate in related models. |
| Cross-Validation Folds | 5 | Folds for hyperparameter selection. |
| Training Epochs | 10 | Epochs for graph neural networks. |
| Budget Evaluations | 5000 | Budget for de novo design evaluations. |
| Population Size (GA) | 250 | Population size for genetic algorithm. |
| Offspring Size (GA) | 25 | Offspring size for genetic algorithm. |
| Mutation Rate (GA) | 0.01 | Mutation rate for genetic algorithm. |
| **liGAN** | | |
| Batch Size | 32 | Batch size for training. |
| Learning Rate | $10^{-4}$ | Learning rate for Adam optimizer. |
| Epochs | 100 | Number of training epochs. |
| Latent Dimension | 256 | Dimension of latent space in GAN. |
| Discriminator Layers | 3 | Number of layers in discriminator. |
| Generator Layers | 4 | Number of layers in generator. |
| **ALIDIFF** | | |
| Batch Size | 4 | Batch size during pretraining. |
| Learning Rate (Pretraining) | 0.001 | Learning rate for Adam optimizer in pretraining. |
| Learning Rate (Fine-tuning) | $5 \times 10^{-6}$ | Initial learning rate for fine-tuning. |
| Adam Betas | $(0.95, 0.999)$ | Beta values for Adam optimizer. |
| Gradient Norm Clipping | 8 | Gradient norm during pretraining. |
| Atom Type Loss Scaling | 100 | Scaling factor for atom type loss. |
| Gaussian Noise Std | 0.1 | Standard deviation for data augmentation. |
| $\beta$ | 5 | Beta value for fine-tuning. |
| **DRUGFLOW** | | |
| Virtual Nodes $N_{\text{max}}$ | 10 | Maximum virtual nodes, remove 5 on average. |
| Sampling Steps | 500 | Number of sampling steps. |
| Training Epochs | 600 | Epochs for DrugFlow and DrugFlow-OOD. |
| $\beta$ (PA) | 100 | Beta for preference alignment. |
| $\lambda_{\text{coord}}, \lambda_{\text{atom}}, \lambda_{\text{bond}}$ | 1, 0.5, 0.5 | Weight for coordinate, atom, bond loss. |
| $\lambda_w, \lambda_l$ | 1, 0.2 | Weight for $w$, $l$ loss. |
| Regularization $\lambda$ (OOD) | 10 | Regularization for uncertainty estimation. |
| Scheduler $k$ (FlexFlow) | 3 | Polynomial scheduler exponent. |
| **DIFFSBDD** | | |
| Batch Size | 32 | Batch size for training. |
| Learning Rate | $10^{-4}$ | Learning rate for Adam optimizer. |
| Diffusion Steps | 1000 | Number of diffusion steps. |
| Number of Layers | 6 | Number of layers in the model. |
| Hidden Size | 128 | Hidden state size. |
| Training Epochs | 100 | Number of training epochs. |

## L.3 MAGICDOCK

Table 12 outlines MagicDock's core hyperparameters, like inversion iterations and loss weights, supporting reproducible optimization for docking-oriented de novo ligand generation.

Table 12: Hyperparameters for MagicDock.

| Hyperparameter | Value | Description |
|---|---|---|
| **MagicDock** | | |
| Hidden units in MLP | 128 | Hidden units in two-layer MLP for pseudo-curvature prediction. |
| Batch size (fine-tuning) | 16 | Batch size during supervised fine-tuning. |
| Learning rate (fine-tuning) | $10^{-4}$ | Learning rate for Adam optimizer in fine-tuning. |
| Weight decay | $10^{-5}$ | Weight decay for Adam optimizer in fine-tuning. |
| Optimizer | Adam | Optimizer used for training. |
| Patch size $K$ | 32 | Patch size for point cloud patches. |
| Max spherical order $L$ | 2 | Max spherical order for SE(3) convolutions. |
| Codebook size $N_B$ | 512 | Codebook size for vector quantization. |
| Gumbel temperature $\tau$ | 1.0 | Temperature for Gumbel-softmax in pre-training. |
| Loss weights $\alpha$, $\beta$ | 5.0, 50.0 | Weights for pocket and interaction losses in composite objective. |
| Loss weight $\lambda_p$ | 1.0 | Weight for delta-G loss in fine-tuning. |
| SDF-GD steps $T_{\text{sdf}}$ | 50 | Number of gradient descent steps in SDF projection. |
| Learning rate $\alpha_{\text{sdf}}$ | 0.1 | Learning rate for SDF gradient descent. |
| Inversion iterations $T$ | 200 | Number of iterative refinement steps in inversion. |
| Step size $\eta_t$ | 0.01 | Step size for gradient updates in inversion stage. |
| Pre-training epochs | 100 | Number of epochs for pre-training stage. |
| Fine-tuning epochs | 20 | Number of epochs for fine-tuning stage. |

