# OpenReview forum: "Toward Protein Docking-oriented De Novo Ligand Design via Gradient Inversion"
_ICLR.cc/2026/Conference — ICLR 2026 Conference Withdrawn Submission_

### Official Review · Reviewer_beRR · 2025-10-27

**Soundness:** 3
**Presentation:** 2
**Contribution:** 3
**Rating:** 6
**Confidence:** 2

**Summary:**

The authors propose **MagicDock**, a docking-based method for *de novo* ligand design. The motivation is to address the limitations of existing methods, namely “Pseudo De Novo,” “Limited Docking Modeling,” and “Inflexible Ligand Type.” The proposed four-stage system demonstrates promising results across several evaluations.

**Strengths:**

- The paper is well-motivated and addresses a relevant problem in *de novo* drug design.
- The authors appear to have conducted comprehensive experiments and theoretical analyses to validate their approach.

**Weaknesses:**

1. **Novelty of Ligand Generation**
   One of the main motivations is to move beyond “Pseudo De Novo” ligand generation. However, the paper does not provide sufficient analysis to demonstrate that MagicDock indeed produces significantly more novel ligands compared to prior methods. For example, a clearer comparison (such as generating antibodies that without given the conserved CDR regions, as mentioned in line 41) would strengthen the claim of novelty.

2. **Details of Inversion-based Generation**
   Since the inversion-based ligand generation mechanism is central to the method’s novelty, Section 3.4 should provide more implementation and methodological details. This would allow readers to better understand and reproduce the proposed approach.

**Questions:**

From line 194, it appears that $P_{\text{lig}} $ has a fixed size  N \and only the point cloud positions are updated. How is N determined during inference? Does the choice of N affect performance, for instance, if N is set too small or too large?

---

### Official Review · Reviewer_qWQw · 2025-10-28

**Soundness:** 2
**Presentation:** 1
**Contribution:** 2
**Rating:** 4
**Confidence:** 4

**Summary:**

This paper introduces a ligand design framework based on docking prediction. It utilizes a predictive model capable of performing docking predictions between molecules and proteins, as well as between proteins themselves. By applying the concept of gradient inversion (i.e., optimization), the model is transformed into an approximate generative model that can iteratively update its inputs.

**Strengths:**

1. This paper unified both ligand types of proteins and ligands, enabling flexible ligand design.
2. The proposed framework unified prediction and generation with one prediction network, which provide broader applicability.

**Weaknesses:**

1. The paper writing needs improvement. This paper is more like a technical report without clear motivation for every component and insight behind the technical design.
2. The framework is way too more complex than unified generative frameworks, without clear ablations for each component.
3. Optimization-based framework is slower than generative framework during inference, which is the major concern for users. Also, the theoretical analysis presented for this framework is overly vague and generic, as the same theoretical principles could be applied to any gradient inversion-based framework.
4. The validation is limited. This paper lacks analysis about the chemical validity and synthesizability, which is usually the major drawbacks for optimization-based method. Meanwhile, the prediction and generation shares the datasets, which may contain biases causing issues like reward hacking.
5. The entire framework is an integration of many existing methods. The optimization-based framework is used for generation, which is not novel even in the biological domain and is commonly applied to predictive networks with AlphaFold-level accuracy. Therefore, the proposed framework lacks innovative design and does not provide new insights. Moreover, the motivation for choosing an optimization-based framework is unclear. In theory, any optimization method could be incorporated into a classifier-guidance or stochastic optimal control framework, which would also allow one to benefit from the likelihood modeling capabilities and inference acceleration techniques offered by generative models.

**Questions:**

1. In Fig.4, the visualization process is a bit confusing. Why the predicted pocket becomes the Generated Ligand? Also there is no conformational change or ligand type change, thus this is not straightforward for visualization.
2. Are the downstream tasks chained during both SFT and Inference? That’s, predict the pocket first, then predict the interaction, and predict the energy change.
3. Will the gradient signal mainly come from the energy prediction head? The signal from a binary interaction prediction is not strong enough for SBDD, it would be interesting to see the contribution ratio for different objectives in the compositional loss.
4. Why VQ in a gradient-inversion framework? This is not common and convenient.

---

### Official Review · Reviewer_YSYY · 2025-10-31

**Soundness:** 2
**Presentation:** 2
**Contribution:** 2
**Rating:** 2
**Confidence:** 3

**Summary:**

This paper proposes MagicDock, a four-stage inversion-based framework for de novo ligand design. The method integrates differentiable surface modeling, SE(3)-equivariant pretraining, supervised fine-tuning on docking-related tasks, and gradient-driven ligand generation. The authors claim that this inversion mechanism allows for end-to-end optimization and generalization across protein and small-molecule ligands. Experiments on datasets such as SKEMPI v2, SAbDab, PDBBind2020, and CrossDocked2020 are reported, showing improved docking metrics over prior approaches.

**Strengths:**

The paper presents a comprehensive engineering pipeline, with well-structured stages and clear modularization.

The formulation of an inversion-based refinement process is technically consistent with recent gradient-based optimization frameworks.

The inclusion of SE(3)-equivariant modeling and surface point cloud representations shows awareness of current practices in protein structure modeling.

Experimental evaluation spans both protein and small-molecule ligands, demonstrating some breadth of applicability.

**Weaknesses:**

Lack of conceptual novelty.
The central idea—gradient-based inversion on differentiable surface embeddings—is not new and has already appeared in recent works such as Boltz-2 (Nature 2024), Chai-2, and AlphaFold3’s structure-diffusion inversion modules, which perform fine-grained energy and docking optimization using gradient-based updates. The proposed “MagicDock” framework appears to repackage existing paradigms (diffusion, flow matching, SE(3)-equivariant GNNs, and gradient descent in latent space) rather than contributing a new methodological insight.

Marginal empirical improvement.
Reported performance gains (e.g., +11.7% on molecular benchmarks) are incremental and within the variance seen across modern baselines. Given that the field is rapidly progressing—with large-scale multimodal foundation models like AlphaFold3, Chai-2, and Boltz-2 capable of full protein–ligand co-folding and docking prediction—the incremental performance increase is not scientifically compelling.

Overstated theoretical contribution.
The claimed “rigorous theoretical guarantees” are generic restatements of standard optimization theory (e.g., convergence of projected gradient descent, SE(3)-equivariance proofs, and information-theoretic upper bounds) without being specialized to docking or ligand design. These sections add mathematical verbosity without offering domain-specific insight or new theorems.

Limited biological validation.
The paper lacks experimental or structural validation (e.g., docking energy correlation with experimental binding affinities or AlphaFold-Multimer structures). Without such validation, the biological significance of the generated ligands remains speculative.

Outdated comparative context.
The authors compare mainly against diffusion- and RL-based generative baselines but omit the latest structure-to-ligand co-design models (e.g., Chai-2, EquiDock++, RoseTTAFold All-Atom, and AlphaFold3 Multimer). These models already achieve joint receptor–ligand modeling with differentiable geometry, making MagicDock’s inversion strategy appear redundant.

Ambiguous scientific impact.
The “gradient inversion” narrative largely reframes standard backpropagation through a pretrained model as if it were a distinct generative mechanism. Without novel inductive biases, new architectures, or validated improvements in docking precision, the work fails to advance understanding or methodology in protein–ligand generative modeling.

**Questions:**

N/A

---

### Official Review · Reviewer_C1qQ · 2025-10-31

**Soundness:** 3
**Presentation:** 2
**Contribution:** 2
**Rating:** 2
**Confidence:** 3

**Summary:**

The authors propose MagicDock, a gradient inversion method for de novo generation of ligands.

**Strengths:**

I appreciate the extensiveness of the pseudocode, theorems, and detail in the main text and appendix.

**Weaknesses:**

See questions section.

I’m not sure about the methodological novelty of this method. “Patching”/masked autoencoding has been applied before for proteins and molecules, and SFT/gradient inversion is not a new method.

I would appreciate error bars on tables 1 and 2. While MagicDock is the best in most cases, it appears only marginally better than previous methods such as DiffSBDD and DrugFlow.

The font on the figures could be made bigger. Right now the figures are too packed with information.

**Questions:**

I’m not sure about the weaknesses you mention in the introduction. Specifically, you claim that prior methods do not explicitly consider “spatial docking information and protein surface information” (intro, 2nd paragraph). Many prior methods do incorporate protein pocket information (DrugFlow, DecompDiff, etc), and information on atom types, charges, hybridization, etc.

What are the specific features in eq 5? The manuscript just says a “highly comprehensive feature vectors”. What are the features?

How are stages 3 and 4 related? Is binding affinity regression SFT done in hopes of improving the ligand generation method?

---

### Note · Authors · 2025-11-16

I have read and agree with the venue's withdrawal policy on behalf of myself and my co-authors.